# Aging alters neural activity at event boundaries in the hippocampus and Posterior Medial network

Zachariah M. Reagh [1,2,3 ✉], Angelique I. Delarazan[1,2], Alexander Garber[2] & Charan Ranganath[2,4]

Recent research has highlighted a role for the hippocampus and a Posterior Medial cortical network in signaling event boundaries. However, little is known about whether or how these neural processes change over the course of healthy aging. Here, 546 cognitively normal participants 18–88 years old viewed a short movie while brain activity was measured using fMRI. The hippocampus and regions of the Posterior Medial network show increased activity at event boundaries, but these boundary-evoked responses decrease with age. Boundary-evoked activity in the posterior hippocampus predicts performance on a separate test of memory for stories, suggesting that hippocampal activity during event segmentation may be a broad indicator of individual differences in episodic memory ability. In contrast, boundary-evoked responses in the medial prefrontal cortex and middle temporal gyrus increase across the age range. These findings suggest that aging may alter neural processes for segmenting and remembering continuous real-world experiences.

[1] Department of Psychological & Brain Sciences, Washington University in St. Louis, St. Louis, MO, USA. [2] UC Davis Center for Neuroscience, University of California, Davis, CA, USA. [3] Department of Neurology, University of California, Davis, CA, USA. [4] Department of Psychology, University of California, Davis, CA, USA. ✉email: zreagh@ucdavis.edu

The process of parsing a continuous stream of information into meaningful chunks—event segmentation—is thought to be a fundamental process that influences event comprehension, episodic memory retrieval, and prediction[1–3]. Segmentation ability correlates with individuals' ability to later remember information, suggesting that event boundaries play a powerful role in organizing information in memory[4,5]. It follows that event segmentation should recruit neural mechanisms involved in the formation of episodic memories. Consistent with this idea, an emerging body of evidence suggests that, in young adults, hippocampal activity is enhanced during perception of boundaries in naturalistic events, and that boundary-evoked activity appears to be related to changes in representational patterns in the neocortex[6,7]. Importantly, the hippocampus is known to interact closely with two large-scale cortical networks: a Posterior Medial (PM) network that is more affiliated with the posterior hippocampus, and an Anterior Temporal (AT) network that is more affiliated with the anterior hippocampus. Available evidence suggests that activity in the PM network may be sensitive to the structure of events[8–10], including event boundaries[6,7].

Event segmentation shows high levels of agreement (i.e., correlated event boundary estimates across participants) in healthy adults[11]. Evidence pertaining to how segmentation agreement changes with age is mixed, with some older adults showing a slight decrease in agreement[12–15] (particularly, in the case of Alzheimer's disease[12]) and others showing no difference[5,16,17] compared with younger adults. Critically, very little is known about how the neural mechanisms of event segmentation change across the lifespan. Given that aging is consistently associated with changes in episodic memory[18], and with changes in hippocampal[19,20] and PM network function[21–23], we hypothesized that aging would disrupt neural responses in these regions at event boundaries. Specifically, the present study aims to answer the following questions: (1) Is boundary-evoked activity in the hippocampus and PM network affected by aging? (2) If so, are these changes linked to episodic memory across individuals?

Here, we examine the relationship between age and neural activity at event boundaries in a subset of the Cambridge Centre for Ageing and Neuroscience (CamCAN) data set ($N = 546$)[24,25]. This very large sample size allows not only for binned comparisons across age groups, but well-powered assessments using age as a continuous variable. Participants underwent functional magnetic resonance imaging (fMRI) scanning as they viewed a shortened 8-minute version of the Alfred Hitchcock film, "Bang! You're Dead". We find that activity evoked by event boundaries in the hippocampus and PM network regions—which has previously been seen in this data set[7]—is significantly disrupted in the aging brain. We further find that boundary-evoked activity correlates with episodic memory performance on a separate neuropsychological test of narrative memory across individuals.

## Results

### Hippocampal boundary-evoked responses decline with age.
Our primary analyses tested the hypothesis that hippocampal activity would be enhanced at event boundaries for younger subjects, and that this effect would be disrupted with increasing age. Although event boundary ratings were not obtained from the individuals in the CamCAN data set, a separate group of individuals provided these ratings (as used in ref. [7]), and only boundaries with at least 50% intersubject agreement were used in our analyses. To enhance the specificity of our estimates of activity at event boundaries, two steps were taken: (1) we modeled a regressor of non-interest modeling high-frequency visual information, (2) we modeled within-event timepoints equal to the number of boundary timepoints (see Methods for details), which we subtracted from

boundary-evoked activity. This subtraction procedure addresses potential ambiguities in whether age-related changes in boundary-evoked activity are specific to event boundaries per se, or alternatively, a change in overall activity levels (i.e., during event boundaries, as well as within events). Our analyses here will thus be conducted over boundary-evoked activity minus within-event activity. However, analyses related to boundary-evoked activity alone (without this adjustment) can be seen in Supplementary Figs. 1–4. We note that the basic findings reported below are unchanged as a function of this adjustment.

Based on their differential connectivity with the PM and ATs networks, we defined separate regions of interest (ROIs) for the posterior (pHPC) and anterior (aHPC) hippocampus (divided longitudinally at the uncal apex). In pHPC, we observed significant boundary-evoked activity ($t_{(545)} = 31.266$, $p = 1.11e^{-123}$) that decreased significantly with age across the sample ($r = -0.345$, $p = 1.06e^{-16}$) (Fig. 1a; see Supplementary Fig. 1 for unadjusted data). Breaking the full range down into three different age groups (an equal split of 182 participants per group), we found a significant difference across groups ($F_{(2,543)} = 31.684$, $p = 9.66e^{-14}$, $\eta^2 = 0.105$) (Fig. 1b). Boundary-evoked activity was higher in Young than middle-aged and older adults, and furthermore, higher in middle-aged adults than in older adults (all $p < 0.05$ corrected).

Boundary-evoked activity was statistically significant in aHPC ($t_{(545)} = 4.94$, $p = 1.54e^{-42}$), but despite considerable statistical power, we found no significant age-related changes either in analyses across the age continuum ($r = -0.059$, $p = 0.16$) (Fig. 1c) or when broken down by age groups ($F_{(2,543)} = 2.55$, $p = 0.105$, $\eta^2 = 0.001$) (Fig. 1d). Importantly, the negative relationship between age and boundary-evoked activity in pHPC was significantly stronger than in aHPC ($z = -4.95$, $p < 0.001$). These results indicate that aging disproportionately attenuates posterior hippocampal activation at event boundaries. Finally, visualizing the data as a time-course of activity averaged across the modeled event boundaries, it is clear that the age-related decline in pHPC reflects a reduction in the amplitude of the hemodynamic response, rather than a fundamentally different shape, or a temporal shift of the response (Fig. 2a). Thus, age-related differences are not owing to a failure to account for a shifted peak of the response. It is also clear that boundary-evoked response time courses and their relationship with age are less robust in aHPC than pHPC (Fig. 2b).

### Hippocampal boundary-evoked responses predict story memory.
Prior studies suggest that individual differences in event segmentation predict memory for complex events[5]. If this effect is related to hippocampal activity at event boundaries, we would expect participants with less boundary-evoked activity to show poorer memory for complex events. Although memory for the video stimulus used during the scan session was not assessed in this experiment, the participants completed a separate test of memory for narratives—the Logical Memory portion of the Wechsler Memory Scale. These stories, like the film used in the fMRI study, consist of a series of thematically linked events. If hippocampal activity at event boundaries is related to individual differences in the ability to remember complex events, then we would expect participants who showed higher boundary-evoked activity to show better memory performance.

Average values for ROI activity at event boundaries, neuropsychological test scores, age, and framewise displacement are displayed across age groups in Table 1. In pHPC, activity at event boundaries predicted recall in the Logical Memory task stories both immediately ($r = 0.157$, $p = 0.0002$) (Fig. 3a; see Supplementary Fig. 2 for unadjusted data), and after a 20-minute delay ($r = 0.209$, $p = 7.26e^{-07}$) (Fig. 3b). This relationship was not

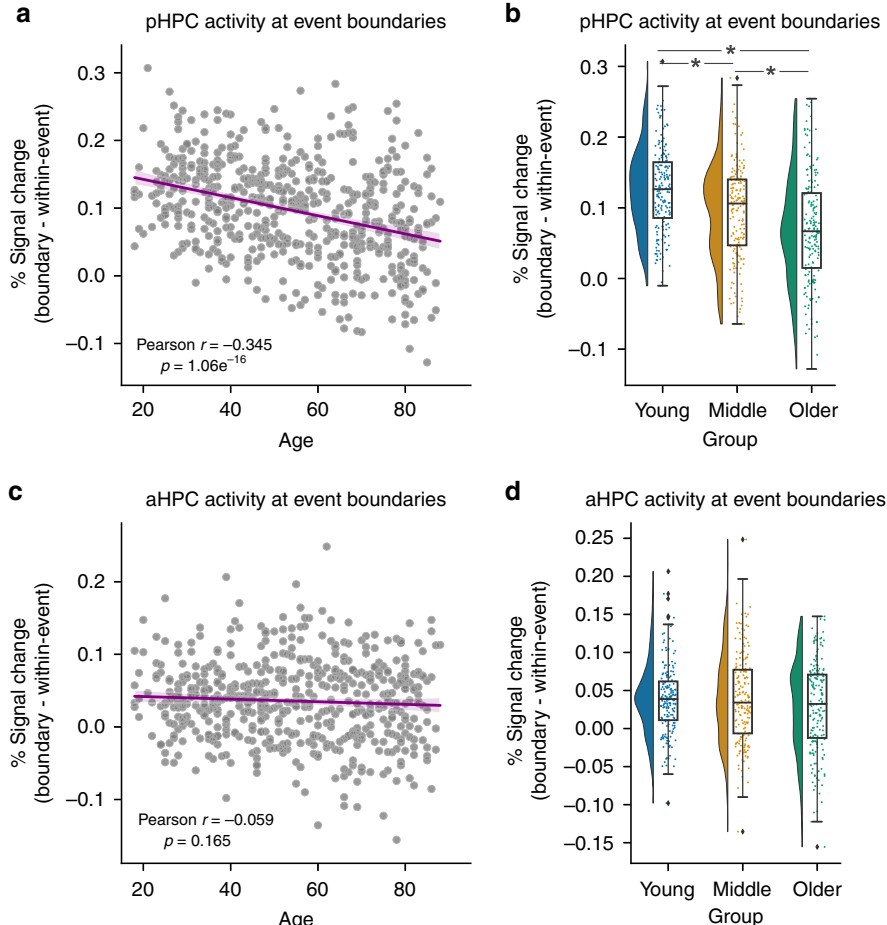

**Fig. 1 Boundary-evoked activity and age-related decline in posterior hippocampus. a** Activity at event boundaries significantly declines with age in pHPC. **b** This effect holds when grouping, and comparing across young, middle, and older individuals. No relationship between age and aHPC boundary-evoked activity was observed (**c**) nor any difference across age groups (**d**). (Correlations were assessed via Person's r. * indicates a significant difference via Tukey's HSD at $p < 0.05$ corrected following a one-way ANOVA. Raincloud plots depict data distributions across groups, with box components displaying median values and data quartiles. $N = 546$ for correlations, $N = 182$ per group for groupwise comparisons, examined over a single experiment).

specific to any age group, as older adults ($r = 0.244$, $p < 0.001$), middle aged ($r = 0.201$, $p = 0.006$) and younger adults ($r = 0.181$, $p = 0.014$) all featured a significant correlation. Though the strength of the correlation appears to increase with age, pairwise comparisons between the strength of these correlations (via Fisher z-transforms) did not reveal significant differences. In contrast, despite considerable statistical power, there were no other ROIs for which activity was predictive of story recall. In addition, only a marginal correlation was observed between boundary-evoked activity in aHPC and logical memory performance at a delay that did not reach significance ($r = 0.071$, $p = 0.096$) (Fig. 3d), and no meaningful correlation between aHPC and immediate recall (Fig. 3c).

Increased age was associated with poorer recall on the logical memory test memory both immediately ($r = -0.293$, $p = 2.61e^{-12}$) and at a delay ($r = -0.324$, $p = 8.64e^{-15}$). Given that boundary-evoked pHPC activity also declined with age, we further explored the relationship between the age, pHPC activity, and logical memory performance. We conducted a multiple linear regression analysis, entering age, boundary-evoked activity from all ROIs, head motion in the scanner, as well as all neuropsychological score variables of interest as predictors of story memory. The only significant predictor of immediate recall on the logical memory test was delayed recall on the same test ($p < 0.001$). For delayed recall, immediate logical memory recall accounted for the most variance ($p < 0.001$), followed

by age ($p = 0.004$). However, pHPC activity accounted for significant variance in delayed story recall over and above what was accounted for by immediate recall performance and age ($p = 0.035$) (Table 2). We next investigated whether individual differences in boundary-evoked pHPC activity was reflective of general individual differences in cognitive ability or even performance on simple item memory tasks. Surprisingly, we did not observe any significant relationships between pHPC boundary-evoked activity and any other neuropsychological test of interest, including composite tests of word memory, verbal fluency, and visuospatial performance (all $p > 0.05$; minimal $p = 0.15$ for word memory).

Putting results from these analyses together, we found that: (1) individual differences in hippocampal activity at event boundaries during viewing of a film predicted the ability to retain information about spoken narratives in an entirely different task context; (2) although pHPC activity at event boundaries was correlated with age, pHPC activity accounted for long-term retention of story information above and beyond the strong effect of age, and activity across other ROIs (Fig. 4); (3) these effects were specific to measures of narrative memory, in the sense that pHPC activity was not significantly correlated with other neuropsychological measures.

**Aging changes boundary-evoked responses in the PM Network.** We next examined boundary-evoked activity in our cortical ROIs

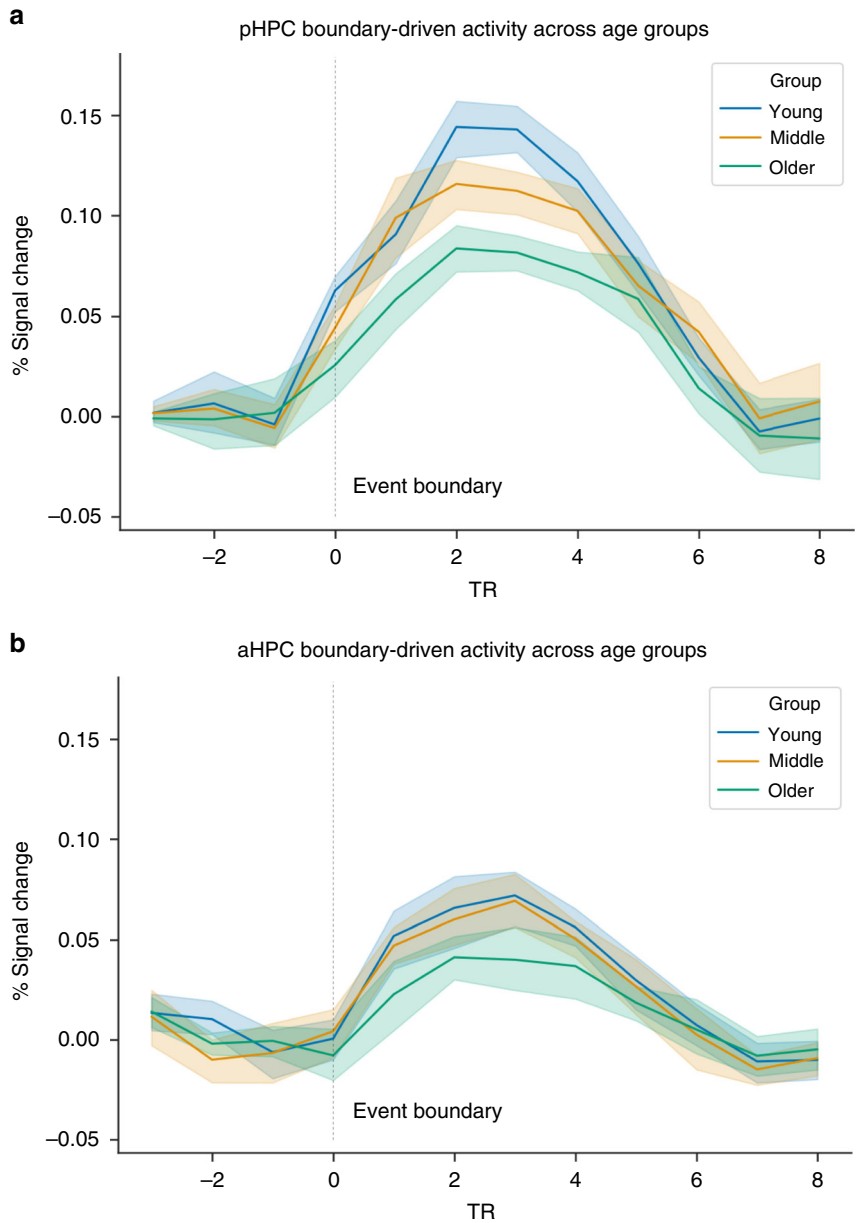

**Fig. 2 Time-course of boundary-evoked response in posterior hippocampus across age groups.** The time-courses of BOLD responses (FIR) are highly comparable across groups, indicating reduced pHPC boundary-evoked responses with age rather than a fundamentally different shape of the hemodynamic response. **a** pHPC, **b** aHPC.

(Fig. 4). Given the preferential connectivity of pHPC to the PM Network[26], we predicted that PM regions, like pHPC, would be driven by event boundaries. Like pHPC, we found significant boundary-evoked activity throughout PM network regions: angular gyrus (ANG; $t_{(545)} = 37.26$, $p = 5.61e^{-152}$), posterior medial cortex (PMC; $t_{(545)} = 32.69$, $p = 5.61e^{-130}$), and parahippocampal cortex (PHC; $t_{(545)} = 27.58$, $p = 5.61e^{-105}$). Also in line with pHPC, we observed significant negative correlations with age in the same regions: ANG ($r = -0.212$, $p = 5.56e^{-07}$), PMC ($r = -0.174$, $p = 4.37e^{-05}$), PHC ($r = -0.131$, $p = 0.002$) (Fig. 5a, c, e). Broken down by groups, we found significant differences in boundary-evoked activity across age groups in each region: ANG ($F(2,543) = 19.564$, $p = 6.24e^{-09}$, $\eta^2 = 0.049$), PMC ($F(2,543) = 8.705$, $p = 0.002$, $\eta^2 = 0.032$), PHC ($F(2,543) = 3.295$, $p = 0.038$, $\eta^2 = 0.014$) (Fig. 5b, d, f). In ANG and PMC, post hoc contrasts revealed that both young and middle-aged adults had significantly higher boundary-evoked activity than Older adults

(all $p < 0.05$ corrected), whereas the effect was driven by the difference between young and older adults in PHC ($p < 0.05$ corrected).

Interestingly, and somewhat surprisingly, not all PM network regions showed declining boundary-evoked activity with age. Like ANG, PMC, and PHC, we observed significant boundary-evoked responses in medial prefrontal cortex (mPFC; $t_{(545)} = 26.31$, $p = 1.33e^{-147}$) and middle temporal gyrus (MTG; $t_{(545)} = 19.47$, $p = 1.68e^{-164}$). In contrast to ANG, PMC, and PHC, we observed increasing activity at event boundaries across the age range in mPFC ($r = 0.173$, $p = 4.59e^{-05}$) and MTG ($r = 0.124$, $p = 0.004$) (Fig. 5g, i). Broken down into age groups, we found a significant effect of age in mPFC ($F(2,543) = 6.776$, $p = 0.001$, $\eta^2 = 0.025$), driven by lower boundary-evoked activity in young and middle-aged adults compared to Older adults (all $p < 0.05$ corrected) (Fig. 5h). Though the three group analysis revealed a similar trend in MTG, it did not reach statistical significance

**Table 1 Set of group averages for predictor variables entered into multiple linear regression analyses.**

| Measure | Young | Middle | Older |
|---|---|---|---|
| Age | 32.78 | 53.967 | 75.505 |
| Motion | 0.087 | 0.104 | 0.096 |
| pHPC activity | 0.146 | 0.118 | 0.087 |
| aHPC activity | 0.059 | 0.058 | 0.046 |
| ANG activity | 0.080 | 0.072 | 0.054 |
| PMC activity | 0.089 | 0.079 | 0.065 |
| PHC activity | 0.042 | 0.039 | 0.033 |
| mPFC activity | 0.067 | 0.067 | 0.082 |
| MTG activity | 0.046 | 0.054 | 0.061 |
| PRC activity | 0.002 | −0.001 | −0.001 |
| TP activity | −0.007 | −0.001 | −0.009 |
| VC activity | 0.016 | 0.005 | 0.01 |
| Log Mem Imm | 16.09 | 14.53 | 13.47 |
| Log Mem Del | 14.67 | 13.78 | 11.41 |
| Verbal fluency | 12.34 | 12.03 | 9.94 |
| Visuospatial | 15.49 | 14.53 | 14.0 |
| Word memory | 23.65 | 22.36 | 19.51 |

$(F(2,543) = 2.619, p = 0.074, \eta^2 = 0.012)$ (Fig. 5j; see Supplementary Fig. 3 for unadjusted data).

The findings above demonstrate that both the hippocampus and regions throughout the PM network showed significant boundary-evoked responses. We next examined regions in the AT network that, in contrast to PM network regions, were not expected to encode event structure. Despite considerable power to detect effects in this sample, we did not observe significant boundary-evoked activation in perirhinal cortex (PRC; $t_{(545)} = 0.131$, $p = 0.896$) or temporopolar cortex (TP; $t_{(545)} = −0.014$, $p = 0.311$). Moreover, we did not observe a significant relationship with age in these regions: PRC ($r = −0.013$, $p = 0.762$) or TP ($r = 0.009$, $p = 0.835$ (Fig. 6a–d; see Supplementary Fig. 4 for unadjusted data). We additionally found boundary-evoked responses in the amygdala (AMY; ($t_{(545)} = 5.133$, $p = 3.97e^{−07}$). However, critically, we did not observe a relationship between AMY activity and age ($r = −0.026$, $p = 0.548$) (Fig. 6g, h). Thus, the age-related differences in activity driven by event boundaries in the PM network are not seen in the AT network, or in visual processing regions.

We additionally included a visual cortex (VC) ROI as a control region, testing whether age-related changes in boundary evoked

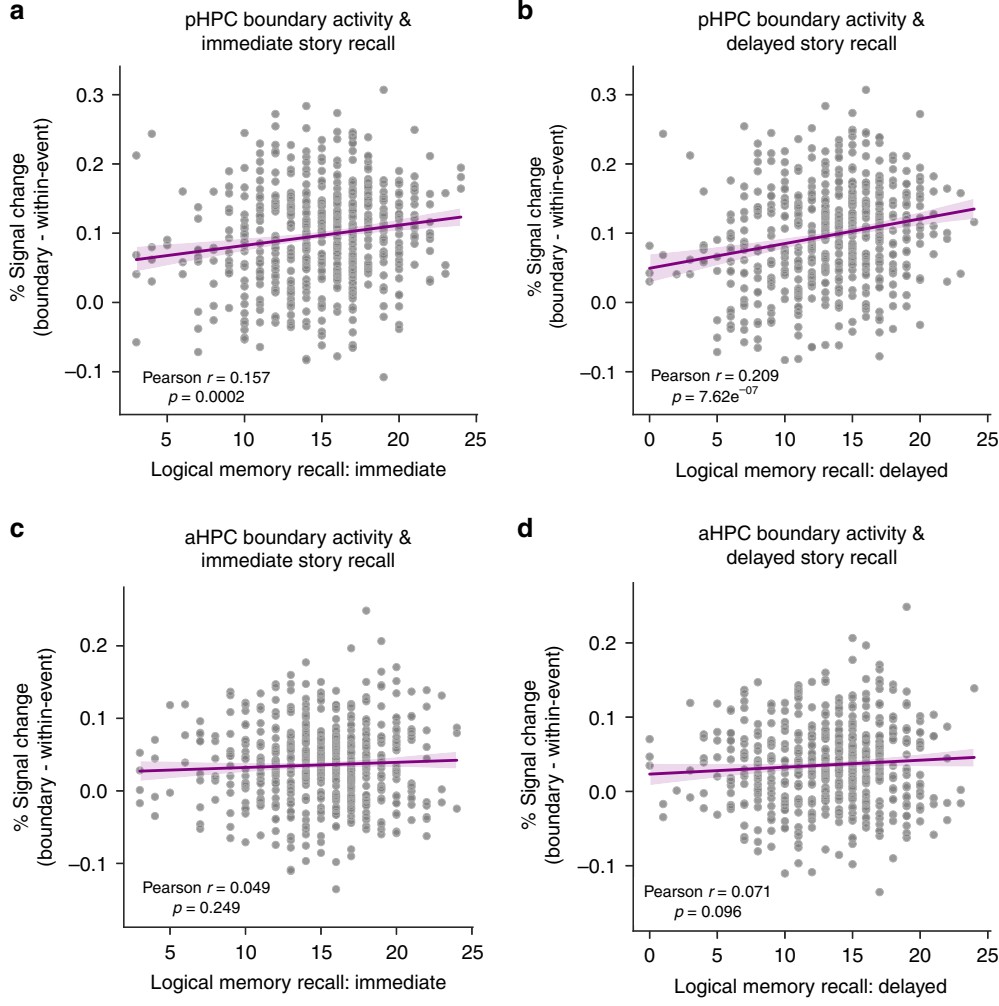

**Fig. 3 Boundary-evoked activity and age-related decline in anterior and posterior hippocampus.** pHPC activity at event boundaries significantly predicts memory for stories in immediate (**a**) and delayed (**b**) recall conditions. These relationships did not reach significance in aHPC (**c**, **d**) (correlations were assessed via Person's *r*).

activity might have been driven by low-level perceptual processes. We did observe significant boundary-evoked activity in VC ($t_{(545)}$ = 4.177, $p = 3.45e^{-05}$), which is perhaps to be expected, as event boundaries in naturalistic stimuli are often associated with visuospatial changes[1–3,17]. However, we did not observe a significant relationship between VC activity and age ($r = -0.055$, $p = 0.209$) (Fig. 6e, f).

**Boundary-evoked activity differs from within-event activity**. We next ran a confirmatory whole-brain voxelwise analysis to assess the selectivity of boundary-evoked responses to the PM network. This voxelwise analysis revealed that boundary-evoked activity, in contrast to within-event activity, was centered on the hippocampus and PM network (false discovery rate $q < 0.05$)

**Table 2 Regression analysis results, predicting delayed logical memory recall.**

| Variable | Coefficient | Std. error | t | p |
|---|---|---|---|---|
| Constant | 0.3505 | 1.048 | 0.334 | 0.738 |
| pHPC | 3.0788 | 1.455 | 2.116 | 0.035* |
| aHPC | 1.9977 | 1.782 | 1.121 | 0.263 |
| PMC | −0.8259 | 1.808 | −0.457 | 0.648 |
| ANG | 0.6639 | 2.124 | 0.313 | 0.755 |
| PHC | 0.6288 | 2.637 | 0.241 | 0.809 |
| MTG | 0.4054 | 1.494 | 0.271 | 0.786 |
| mPFC | −0.6881 | 2.175 | −0.316 | 0.752 |
| VC | −2.3315 | 1.430 | −1.630 | 0.104 |
| PRC | −0.0061 | 1.677 | −0.004 | 0.997 |
| TP | 0.0957 | 0.808 | 0.118 | 0.906 |
| AMY | −0.9888 | 1.583 | −0.625 | 0.532 |
| Motion | 1.5441 | 1.578 | 0.978 | 0.328 |
| Verbal fluency | −0.0107 | 0.045 | −0.237 | 0.812 |
| Visuospatial | 0.0302 | 0.041 | 0.744 | 0.457 |
| Word memory | 0.0033 | 0.026 | 0.126 | 0.900 |
| Log Mem mm | 0.8831 | 0.027 | 33.250 | <0.001* |
| Age | −0.0194 | 0.007 | −2.905 | 0.004* |

(Fig. 7). This result consistent with our ROI-based findings described above, as well as with prior reports indicating BOLD activity in PM regions is sensitive to naturalistic event structure[6,7,27]. Taken together, this confirms that our hippocampal and PM network ROIs are reliably driven by event boundaries, which stands in contrast to activity observed within an event.

**Similar behavioral event segmentation across age groups**. One possible explanation for age-related changes in boundary-evoked responses is inconsistency of perceived event boundaries in older adults. Though we cannot rule out that older participants in the CamCAN data set simply missed a number of the 12 event boundaries we modeled, we addressed this possibility in a separate sample of participants. We collected a sample of 14 older adults (mean age = 73.83 years, SD = 6.27) and 14 younger adults (mean age = 20.15 years, SD = 2.58) to provide some evidence as to whether there were age-associated differences in boundary detection. We did not find evidence for any such differences. In our smaller sample of older adults, at least half of the sample identified each of the 12 boundaries included in our fMRI analyses (i.e., the criterion for the 12 maximal agreement boundaries from the sample of young adults who provided the initial ratings), and the boundaries were overall identified by our sample 86% of the time (compared with 82% of the time in our new young adult sample). Moreover, there were no differences between groups in segmentation agreement among older adults ($r = 0.69$) and among younger adults ($r = 0.73$) ($z = 0.19$, $p = 0.85$), or between either group and agreement across younger and older adults ($r = 0.65$) (young–young vs young–old: $z = 0.36$, $p = 0.719$; old–old vs young–old: $z = 0.17$, $p = 0.87$). Thus, though it cannot be absolutely ruled out, it is unlikely that older adults simply perceived the key event boundaries differently from young adults.

**Discussion**
In the present study, we sought to investigate the effects of aging on neural responses associated with event segmentation.

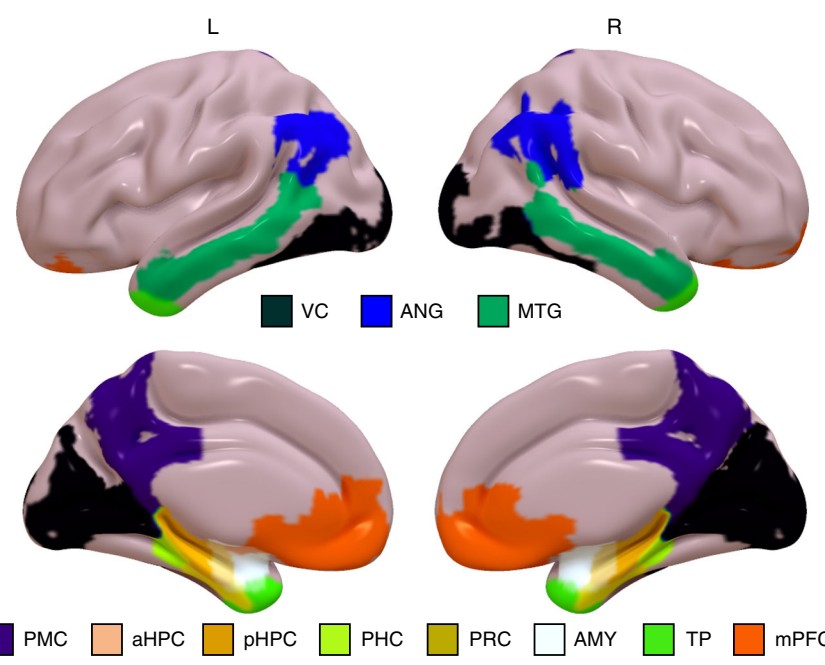

**Fig. 4 Regions of Interest (ROIs).** ROIs are displayed as surface maps normalized to MNI space. VC = visual cortex, ANG = angular gyrus, MTG = middle temporal gyrus, PMC = posterior medial cortex, HPC = hippocampus (anterior/posterior subdivisions not displayed here), PHC = parahippocampal cortex, PRC = perirhinal cortex, AMY = amygdala, TP = temporal pole, mPFC = medial prefrontal cortex).

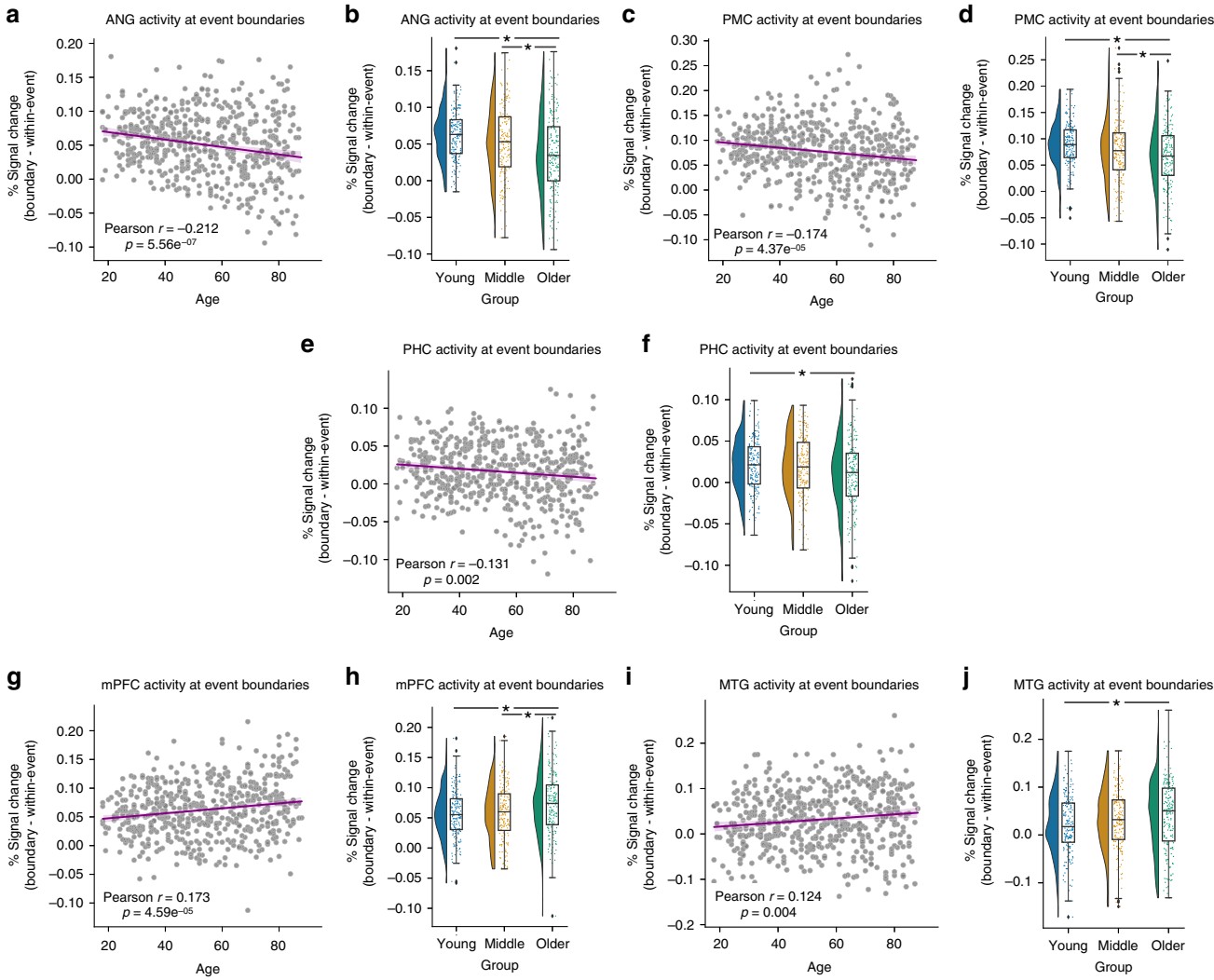

**Fig. 5 Declines and increases in boundary-evoked activity in the Posterior Medial network.** Age-related declines in boundary-evoked responses in angular gyrus (**a**), **b**, posterior medial cortex **c**, **d**, and parahippocampal cortex **e**, **f**. Age-related increases in boundary-evoked responses in medial prefrontal cortex **g**, **h** and middle temporal gyrus **i**, **j**. (Correlations were assessed via Person's $r$. * indicates a significant difference via Tukey's HSD at $p < 0.05$ corrected following a one-way ANOVA. Raincloud plots depict data distributions across groups, with box components displaying median values and data quartiles. $N = 546$ for correlations, $N = 182$ per group for groupwise comparisons, examined over a single experiment).

Hippocampal activity was reliably increased at event boundaries, but we found a dissociation along the longitudinal axis of the hippocampus, such that boundary-evoked responses were larger in pHPC than aHPC. Boundary-evoked activation significantly declined with age in pHPC, and the age-related decline was significantly larger in pHPC than in aHPC. The sensitivity of pHPC activity to event boundaries was behaviorally relevant, in that pHPC activity uniquely predicted memory for narrative information. In addition to pHPC, an affiliated network of neocortical regions showed age-related changes in boundary-evoked activation, with parietal and parahippocampal regions showing decreases and prefrontal and middle temporal regions showing increases with age. Our findings establish a significant role for the hippocampus in neural event segmentation, and they suggest that changes in the pHPC and affiliated neocortical regions may be a sensitive marker of age-related change in the ability to process and remember complex events.

According to Event Segmentation Theory (EST)[1–3], people construct a mental representation of a complex event (i.e., an event model), to comprehend what is happening at the moment and to predict what will happen in the near future. EST predicts that event boundaries occur at points of prediction errors, such that the current event model is abandoned, and a new event model is generated. Many theories have proposed a role for the hippocampus in signaling prediction errors, and there is abundant evidence consistent with this idea[28–30]. Although early fMRI studies of naturalistic stimuli did not reveal a role for the hippocampus in event segmentation[31], recent work—including an analysis of data from young participants in the CamCAN data set[7]—has shown that hippocampal activation is reliably increased at event boundaries. Studies by Baldassano et al.[6] and Ben-Yakov and Henson[7] found that hippocampal activity was enhanced during changes in activity patterns in the PM network that occurred at event boundaries. In the present data, boundary-evoked responses were larger in pHPC, relative to aHPC, fitting with other studies showing that PM cortical regions are more closely affiliated with pHPC than aHPC[26].

The paradigm used in the CAM–CAN data set provides limited insight into the precise factors that triggered activation at event boundaries. Our analyses suggest that low-level visual information does not account for PM network activity at event boundaries, or changes with age. In terms of higher-order factors, it is

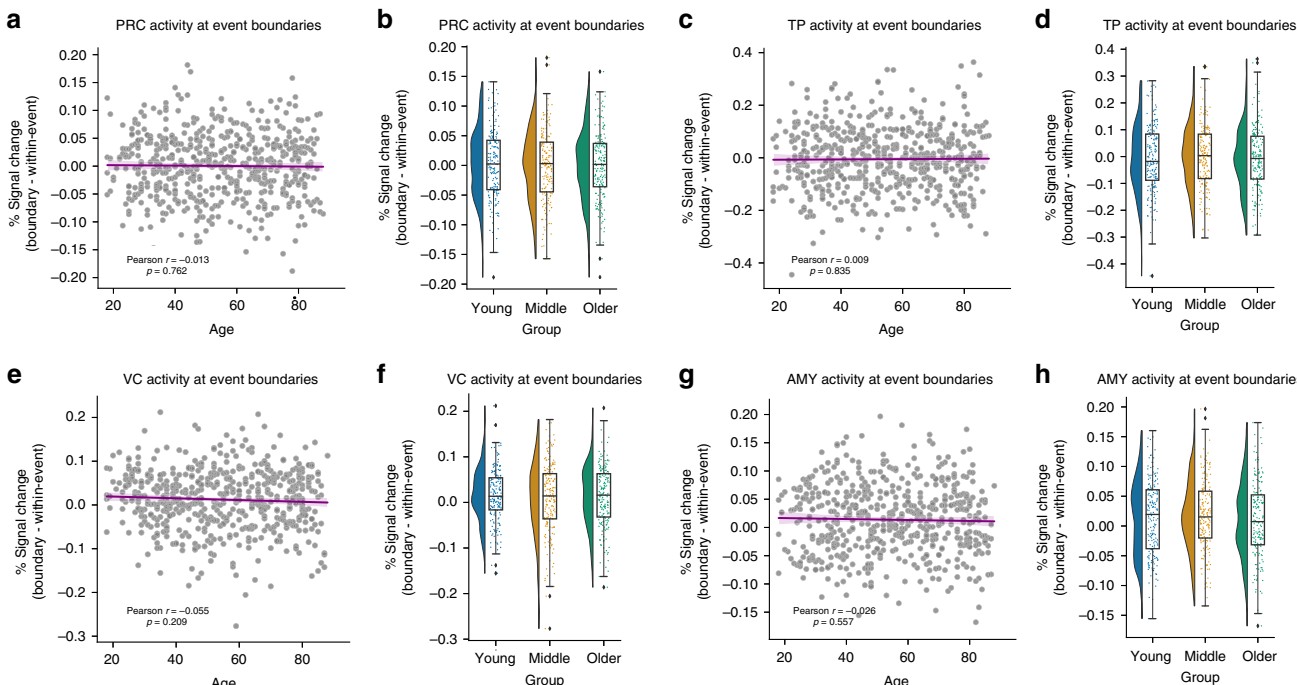

**Fig. 6 Anterior-temporal regions and visual cortex do not show age-related changes.** Boundary-evoked responses were not observed in **a**, **b** perirhinal cortex or **c**, **d** temporal poles. Boundary-evoked responses were observed in **e**, **f** visual cortex and **g**, **h** amygdala, but these did not significantly change with age. (Correlations were assessed via Person's $r$. Raincloud plots depict data distributions across groups, with box components displaying median values and data quartiles. $N = 546$ for correlations, $N = 182$ per group for groupwise comparisons, examined over a single experiment).

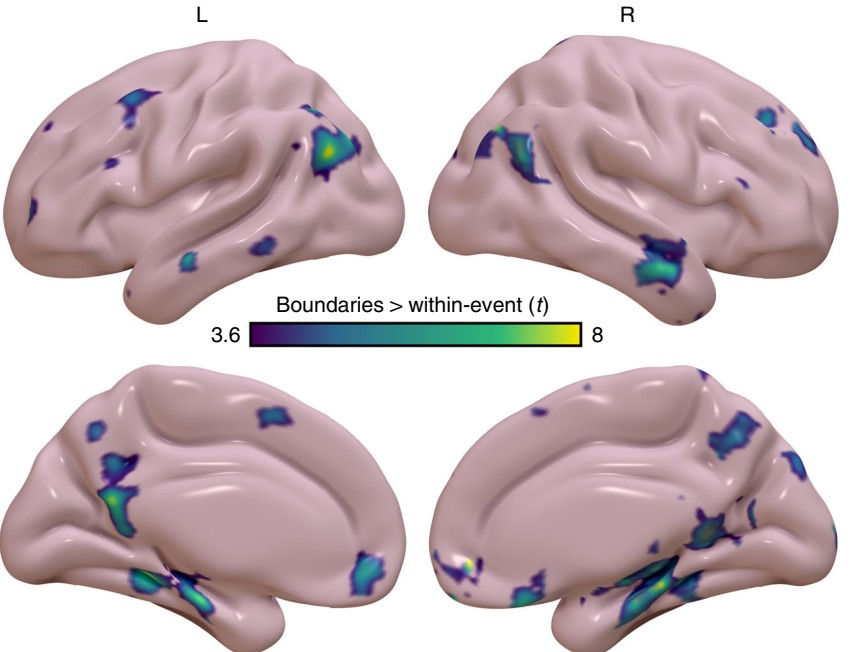

**Fig. 7 Voxelwise analysis of boundary-evoked activity versus within-event activity.** Greater boundary-evoked activity than within-event activity is observed throughout the hippocampus and PM network, consistent with ROI-based analyses. Significance is evaluated at a corrected FDR threshold of $q <$ 0.05.

known that spatial context changes reliably trigger event segmentation[32], and many (but not all) of the boundaries modeled in this 8-minute data set involved a significant change in spatial context (9 out of 12). However, it is well established that event segmentation can also be triggered by prediction errors even in the absence of significant changes in spatial context or sensory input[33] (Zacks et al.[33]). Although we do not have enough event

boundaries in the present data to separately examine activity during spatial context changes and activity during nonspatial changes or prediction errors, this distinction can be fruitfully explored in future studies.

Analyses of age-related changes in event boundary activation revealed that aging disproportionately affected activity in pHPC, relative to aHPC. Importantly, this result in pHPC is

meaningfully linked to memory ability. Of the ROIs investigated here, pHPC activity at event boundaries uniquely predicted delayed recall performance on an entirely separate test of memory for complex narratives. Although age and performance on the corresponding immediate recall test predicted the majority of variance in delayed recall, pHPC activity during event boundaries in the video still predicted significant variance in memory performance. This result is consistent with theories suggesting that event boundaries play a critical role in organizing and later remembering our experiences[1–5,34–36]. Our results point to a role of pHPC responses to event boundaries as a potentially important aspect of this process. Moreover, pHPC activation during event boundaries may have potential as a biomarker for individual differences in the ability to remember complex episodes that consist of multiple events. It is reasonable to think that this biomarker would be more predictive of real-world cognition than the majority of current paradigms.

Outside of the hippocampus, BOLD responses at event boundaries did not uniformly decline as a function of age in the sample. Instead, we found a more complex pattern of results. Whereas PMC, ANG, and PHC showed significant decline in activity with age, activity in mPFC and MTG increased with age. The opposing results across the two sets of ROIs is strong evidence against the possibility that age-related changes simply reflected increased noise, or other nonspecific factors that reduced overall sensitivity of the BOLD signal[37]. This further argues against the possibility that relevant event boundaries were missed in older adults. This dissociation between relatively posterior and relatively anterior PM network subregions aligns with other findings showing that older adults feature reduced activation in posterior cortical areas and increased activation in anterior regions. Some have proposed that this posterior-anterior shift[38] might reflect a process by which the aging brain compensates for reductions in the integrity of posterior regions through increased recruitment of complementary, alternative neural circuits in supporting cognitive operations that might otherwise be weakened or lost[39]. Some theories propose a gradient of cortico-hippocampal representations, with more fine-grained representations in posterior regions and coarser or gist-based representations in anterior regions[40]. Although the present results do not speak directly to this phenomenon, it is possible that they reflect increasing reliance on gist-based representations of events in older subjects.

Although aging is associated with reductions in cognitive control that depend on dorsal fronto-parietal networks, our exploratory voxel-based analyses did not reveal boundary-evoked activation in these regions. In an analysis of the same data set used here, Campbell and colleagues[41,42] report that intersubject correlations (ISC) in brain activity timecourses in these fronto-parietal regions significantly declined with over the adult lifespan. Because ISC measures across-subject consistency in neural responses during viewing of the movie, results of that study suggest that older adults tended to show more idiosyncratic patterns of brain activity. Further analyses revealed that idiosyncratic brain activity timecourses during movie viewing were associated with poorer measures of attentional control. Putting their findings together with results from the present study, it is possible that effects of aging on attentional control and event segmentation might reflect different underlying mechanisms.

The present study examined neural event segmentation in the context of older adults without a dementia diagnosis. However, these results are relevant to the distribution of Alzheimer's disease pathology. Alzheimer's disease is associated with accumulation of amyloid plaques[43–45] and abnormal activity throughout the PM network[46–49]. At present, amyloid pathology in the absence of frank cognitive deficits is a key biomarker defining preclinical Alzheimer's disease[50]. However, this poses a dilemma: as Alzheimer's-related pathology is thought to accumulate years before cognitive symptoms, current assessments could fail to identify those at highest risk during the critical window when treatments and interventions would be most effective. A recent study by Jones et al.[51] suggests that dysfunction in the more posterior aspect of the PM Network occur early in disease progression, and later cascade into more anterior brain networks. This distinction could explain the dissociation between posterior and more anterior PM Network regions in the present data. Critically, the PM Network dysfunction reported by Jones et al. also precedes amyloid accumulation, suggesting that regional dysfunction may be among the earliest possible biomarkers. PM Network functions such as perceiving and remembering lifelike scenes may thus be selectively vulnerable in certain aging individuals, particularly so with amyloid accumulation, as is reported by Maass and colleagues[52]. We suggest that behavioral and neural measures related to complex, naturalistic events may reflect a process that taps into PM network integrity, and our results show that they are sensitive to aging. Importantly, these functional measures relate to processing of lifelike situations, and thus may reflect cognitive status in a way that mirrors daily functioning more closely than the majority of neuropsychological tests. Accordingly, studies of event cognition and event representation may be a promising direction for characterizing preclinical Alzheimer's disease.

## Methods

**Participants**. Participants were drawn from the CamCAN data set (http://www.cam-can.org/index.php?content=dataset). The sample consisted of 623 participants (316 female) for whom all fMRI and neuropsychological data of interest was present. From this initial sample, 546 participants (271 female, mean age = 54.08, SD = 18.56, range = 18–88) were included in our analyses owing to issues with neuroimaging data quality (see Neuroimaging data preprocessing for details). All participants were native English speakers. Given the relatively large sample size, we did not exclude participants based on handedness. Written informed consent was obtained in accordance with the Cambridgeshire Research Ethics Committee.

**Stimuli**. Participants underwent fMRI scanning as they viewed an audiovisual stimulus, a shortened version of Alfred Hitchcock's film "Bang! You're Dead". The stimulus was shortened from its original length of 25-minutes (as in ref. [7,27]) as part of the full CamCAN scanning protocol, though critical narrative elements of the shortened 8-minute movie remain intact[24]. This data set has been used in a number of other fMRI analyses in recent years. Participants were tasked with attending to the movie, and most participants reported having not seen it previously.

Additional measures of interest included performance indices on select neuropsychological tests from the full battery included in the CamCAN protocol. Of particular interest to us were the Logical Memory immediate and delayed recall tests from the Wechsler Memory Scale, and the composite memory, fluency, and visuospatial performance scores from the Addenbrooke's Cognitive Examination (ACE-M; ACE-F; ACE-VS).

**Neuroimaging data acquisition**. Data were collected as a part of a larger scanning protocol in a 3T Siemens TIM Trio, with a 32-channel head coil. The scanning session consisted of structural scans, a resting state scan, and a block of tasks which included movie viewing. High-resolution T1-weighted magnetization Prepared Rapid Gradient Echo (MPRAGE) images were acquired with the following parameters: repetition time (TR) = 2250 ms; echo time (TE) = 2.99 ms; inversion time (TI) = 900 ms; flip angle = 9°; field of view (FOV) = 256 × 240 × 192 mm; voxel size = 1 mm isotropic; GRAPPA acceleration factor = 2; acquisition time = 4 min, 32 s. Functional data during movie viewing consisted of a T2*−weighted Echo Planar Imaging (EPI) sequence with the following parameters: TR = 2470 ms; TE (five echoes) = 9.4 ms, 21.2 ms, 33 ms, 45 ms, 57 ms; flip angle = 78 degrees; 32 axial slices; slice thickness = 3.7 mm with an interslice gap of 0.74 mm (20% of slice thickness); FOV = 192 × 192 mm; voxel size = 3 × 3 × 4.44 mm; acquisition time = 8 min, 13 sec. We additionally incorporated a T2* EPI scan during rest with the same acquisition parameters described above, except the scan duration was 5 min.

**Neuroimaging data preprocessing**. Data were preprocessed using AFNI (version 18.2.15; https://afni.nimh.nih.gov), ANTs (http://stnava.github.io/ANTs). AFNI preprocessing used the standardized afni_proc.py pipeline, with specific steps as follows: (1) despiking of the functional time series (3dDespike); (2) slice timing

correction (3dTshift); (3) coregistration of functional to anatomical images (align_epi_anat.py); (4) motion correction with alignment to the minimum outlier in the time series (3dvolreg); (5) masking of the functional time series to brain voxels as defined by the anatomical image (3dcalc); (6) generation of tissue maps (gray matter, white matter, cerebrospinal fluid); (7) normalization and scaling of each voxel time series and conversion to percent signal change (3dTstat, 3dcalc). We used ANTs to create a group template constructed from all 546 participants (antsMultivariateTemplateConstruction2.sh), and all analyses took place at the level of the group template (in MNI space). Participants whose average framewise displacement exceeded 0.5 mm, whose maximal framewise displacement exceeded 3 mm (derived during motion correction), or whose global temporal signal-to-noise ratio fell two standard deviations below the group mean were excluded from analyses (77 in total).

**Region-of-interest definition.** Given the engagement of the hippocampus at event boundaries[13,14], we included ROI masks for the pHPC and anterior hippocampus (aHPC), as well as AT and PM structures in the medial temporal lobe (MTL): PRC, and PHC. Masks for these regions were adapted from a previous study by Ritchey and colleagues[53]. The aHPC mask consisted only of the hippocampal head (i.e., anterior to the uncus), whereas the pHPC mask consisted of the combined body and tail subdivisions. We used FreeSurfer (https://surfer.nmr.mgh.harvard.edu/) to create a cortical parcellation on the template image (recon-all). We selected specific parcels (Desikan atlas[54]), which corresponded to regions previously shown to be sensitive to event structure: PMC consisted of labels for isthmus and precuneus; ANG; MTG; mPFC consisted of labels for medial orbitofrontal, frontal polar, and rostral anterior cingulate cortex. We additionally included TP and AMY ROIs, which together with PRC, comprise three anterior-temporal (AT) network control regions. We finally included a broad VC ROI for control analyses, which consisted of FreeSurfer labels for lateral occipital cortex, lingual gyrus, and cuneus. ROIs are visualized on the group anatomical template in Fig. 1.

**Neuroimaging data analysis.** Analyses were conducted using AFNI, ANTs, and Python. We estimated the neural response at event boundaries by fitting a general linear model (GLM) over the movie viewing time series. The GLM included a discrete regressor for human-labeled event boundaries, defined by a sample of 16 independent observers gathered by Ben Yakov and Henson[7]. In brief, these event boundaries were gathered by participants who, when watching the eight-minute video stimulus, were simply instructed to press a key when one meaningful unit (i.e., an event) ended and another began, in line with prior studies[5,13,15]. We included boundary timepoints in which at least half of the participants from the sample agreed (within 5 s of a group-meaned boundary time), and which were no closer than 6 s to one another, totaling 12 event boundaries across the time series. The logic for this selection process was to include timepoints that we were confident featured what the majority of observers would perceive as an event boundary. We additionally included an equal number of within-event timepoints to be contrasted with event boundaries. The within-event regressor consisting of 12 timepoints determined by calculating the average elapsed time between event boundaries, and distributing them evenly throughout the video (ensuring that they fell within an event, no >6 sec from an event boundary). We attempted to control for low-level visual information in our GLM by modeling edge pixels in the video (also similar to Ben Yakov and Henson[7]), and entering the proportion of edge pixels in each time point as a continuous regressor. Edge pixels were calculated via a python routine, which read in the video stimulus, split it into its constituent frames, and in an automated fashion performed edge-detection on each frame (using python package opencv). The proportion of edge pixels to total pixel count was calculated (NumPy) for each frame, and was output into a comma-separated value file. We then resampled frame-by-frame edge information to correspond to the temporal scale of the fMRI data by averaging across adjacent frames within the interval of each TR (2470 ms). This resultant temporally smoothed vector served as our estimate of low-level visual information in each timepoint of the video. This approximates an envelope of framewise high-frequency visual information in the video that is modeled independently of event boundaries. This high-frequency visual information vector was entered into the GLM as a regressor of non-interest. Twelve continuous nuisance regressors were also included to account for motion (six motion regressors—x, y, z, pitch, roll, yaw—plus the derivative of each). These 13 nuisance regressors served as the model baseline. We also included a nuisance regressor to account for linear drift. The resultant GLM produced two beta images of interest: (1) one corresponding neural responses at event boundaries, but not within-event, and (2) one corresponding to neural signals that were present during within-event timepoints, but not event boundaries. Our hypotheses were specifically about the former condition. Thus, to control for within-event activity, we subtracted within-event betas from boundary-evoked betas as is displayed in the main figures. Analyses over boundary-evoked activity not featuring this subtraction are available in Supplementary Information. We note that, importantly, our basic findings are unchanged regardless of whether the within-event beta subtraction is performed.

Each participant's anatomical image was warped to the group average template (antsRegistrationSyN), and functional data as well as tissue masks were brought into template space using the transformations applied during anatomical coregistration (antsApplyTransforms). For each participant, a basic quality check was included to ensure acceptable registration was achieved. This check consisted of comparing each participant's gray matter mask to the gray matter mask of the group template, ensuring that no >10% of a given participant's gray matter voxels fell outside the gray matter mask of the group template (3dcalc).

To quantify boundary-evoked activity, we conducted an ROI-based analysis over event boundary response estimates resulting from the GLM. Data were analyzed in two ways: (1) correlations between activity in each region and age across participants, and (2) comparisons between binned age groups. Participants were binned into three age ranges of equal size (N = 182 each): young (mean age = 32.78, range = 18–44), middle (mean age = 53.97, range = 4–65), and older (mean age = 75.51, range = 65–88) groups. These analyses were conducted in Python using the stats and statsmodels packages. For all regions, we tested for (pairwise $t$ tests) and did not observe any significant differences between males and females. Thus, sex differences were not further explored. We additionally conducted a confirmatory voxelwise analysis to examine whole-brain maps of activity corresponding to event boundaries (3dttest + + in AFNI). Data were masked to the extent of brain voxels (but not masked exclusively to gray matter). In line with our central hypotheses, we conducted a direct contrast of boundary-evoked activity minus within-event activity. Voxels featuring significantly greater activity during event boundaries than within-event timepoints were plotted, false discovery rate corrected such that significance was defined as $q < 0.05$.

**Behavioral data.** Behavioral measures included the visuospatial, memory, and verbal fluency components of the Addenbrooke's Cognitive Examination (ACE), as well as the Logical Memory component of the Wechsler Memory Scale. The ACE scores are composite values from a wide-ranging, standardized cognitive evaluation[55], and the Logical Memory scores indicate narrative memory under a free recall condition[56]. Points are earned by recalling individual elements of the stories. Participants were asked to recall the stories immediately post learning, and after a delay of ~20 minutes. Given previous links between the PM network and naturalistic event structure, we were particularly interested in the relationship between boundary-evoked activity and the Logical Memory test, as it taxed recall of stories in a naturalistic format.

Original event boundary ratings were provided by Ben-Yakov & Henson[7], which were acquired using PsychoPy. The same procedure was repeated for an in-house cohort of older and younger participants, addressing possible concerns about age-related differences in event perception. In our sample, 14 older adults (mean age = 73.83 years, SD = 6.27) and 14 younger participants (mean age = 20.15 years, SD = 2.58) were recruited, and gave informed written consent in accordance with the UC Davis Institutional Review Board. Participants were seated at a computer workspace in the lab, viewing the video as presented via PsychoPy, and were instructed to press the Space key any time they felt that one meaningful event ended and another began.

**Multiple linear regression analysis.** To simultaneously assess the predictive power of all predictor variables of interest in explaining variance in story memory, we fit multiple linear regression models (linear_model.LinearRegression in scikit-learn's linear_model package) using an ordinary least squares approach. In brief, this approach allows us to assess the relationship between variables of interest (e.g., posterior hippocampus (pHPC) activity and logical memory delayed recall) while also accounting for other important variables of interest (e.g., age or head motion). The model predicted logical memory performance immediately and at a delay, including all ROIs and neuropsychological test scores, as well as age and head motion (average framewise displacement) as predictor variables.

**Statistical analyses.** All statistical tests were two-tailed and parametric. Relationships between BOLD univariate activity and age, as well as those between BOLD univariate activity and behavioral data were assessed with Pearson correlations. Comparisons across age groups were conducted over each ROI with one-way analysis of variances and post hoc tests were conducted using Tukey's HSD (honestly significant difference). Statistical tests were thresholded at a significance value of $p < 0.05$, unless otherwise noted.

**Reporting summary.** Further information on research design is available in the Nature Research Reporting Summary linked to this article.

## Data availability

To apply for access to raw behavioral and MRI data, a request must be made to the CamCAN group at https://camcan-archive.mrc-cbu.cam.ac.uk/dataaccess/. Analytical code and data for analyses producing the main figures are available at https://github.com/zreagh/NComms_CamCAN. Source data are provided as a Source Data file.

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

## Acknowledgements

We thank the Cambridge Centre for Ageing and Neuroscience, and particularly Darren Price for assistance with and access to the data set. We thank Aya Ben-Yakov and Rik Henson for helpful input, and for providing behavioral event boundary ratings. We thank Derek Huffman, Brendan Cohn-Sheehy, Alex Barnett, Jordan Crivelli-Decker, and Andy Yonelinas for helpful discussions. We thank our funding sources contributing to this work: NIA 1R03AG063224-01 to C.R., ONR Grant N00014-15-1-0033 to C.R., NIA T32AG050061 to Z.M.R., and UC Davis Alzheimer's Disease Center Pilot Grant to Z.M.R. and C.R.

## Author contributions

Z.M.R. processed and analyzed the data. A.I.D. and A.G. collected behavioral event boundary data and contributed to analyses. Z.M.R. and C.R. interpreted the results and wrote the paper.

## Competing interests

The authors declare no competing interests.
