## [Peer Review File · Nature Communications]

Reviewers' comments:

Reviewer #1 (Remarks to the Author):

This manuscript reports on a study of aging in which 546 participants over a large age range watched a short film while their brain activity was recorded using fMRI. The variable of interest was regional activity at the time of event boundaries (defined using a separate group of subjects). The authors report increased hippocampal activity at the time of these event boundaries, but only the posterior hippocampus showed a reduction in this activation as a function of age. This reduction was related to memory for stories on the WMS logical memory subtest. Boundary event-related activation increases were also seen in components of the posteromedial (PM) network, specifically angular gyrus, posteromedial cortex, and parahippocampal cortex and these also showed reduced activation with advancing age. Medial PFC and medial temporal gyrus showed similar activation but increases with age. There was no boundary evoked activity in the anterior temporal (AT) network and consequently no relationship between activation and age. These ROI-based findings were confirmed using voxelwise approaches and, finally, correlations between angular gyrus and hippocampus were seen in younger individuals but these also declined with age.

Overall, I found this manuscript very enjoyable and even exciting to read. The writing is clear and the manuscript is illustrated nicely. The authors tackle a very interesting question, and they bring together both cognitive and neuroscience approaches to answer questions of translational neuroscience interest. The findings are undoubtedly novel and would be of interest to many people working in the fields of both cognitive aging and memory. I have no substantive concerns about methodological factors.

I do have a few questions that arose while reading the manuscript and I think the authors might want to consider these points which I am listing in order of importance:

1. I realize that this is one of several reports implicating posterior brain regions, parts of the PM network, in activation during event boundaries. However, I wonder how much this association is driven by the nature of the stimuli as much as by the "boundary detection" aspect of the task. The PM network shows functional specialization for spatial information, and boundary events, particularly in films (which is the main sort of stimulus used in these studies), are inherently spatial – different events in films are, after all, usually demarcated as different "scenes". I realize that these two concepts are so confounded that they may be impossible to differentiate experimentally (even written or auditory stimuli are likely to induce imagery). And I am definitely not asking the authors to perform another experiment to resolve this. And the authors addressed the perceptual aspects of the task, so it should be clear that is not what I'm talking about. However, at the very least it would be interesting to hear the authors' views on whether the localization of these findings to PM reflect the event-detection aspect of the task, or the spatial aspect of the task, or whether this is impossible to resolve. And as a related question, do the authors have any measure of how often a boundary event coincided with a scene change (as opposed, perhaps to an "action change")?

2. I found the observations of age-related boundary activity in posterior components of the PM network to be interesting, and also consistent with other reports. One new finding is the relationship between posterior hippocampus and memory performance on a narrative task unrelated to the fMRI experiment. Did the authors test for this association in angular gyrus, posteromedial cortex, and parahippocampal cortex? And is there any relationship between the strength of the relationship between posterior hippocampal signal change and angular gyrus signal change and whether angular gyrus is related to memory performance? In other words, I would expect that if the PM regions are highly correlated to posterior hippocampus, they too would be related to memory function, while if this

correlation is lost, they might not be.

3. I think the voxelwise analysis makes the participation of the AT network in boundary event detection unlikely, and I liked the use of AT ROIs as controls. But these AT ROIs were generally small and at the edges of brain, so measurement could be a bit difficult. Could the authors test larger ROIs, particularly amygdala, in their suite of ROIs shown in figure 6?

4. I'm not clear how the authors assessed significance in figure 8. The proper way is an interaction. Did they test a correlation x age interaction?

5. There are a number of papers in the amyloid and tau literature related to "pathological aging" that the authors might wish to read and include. Jones et al (Brain 2015) defined a course of network failure beginning posteriorly and spreading anteriorly, providing a more meaningful explanation of the "posterior-anterior" shift. Maass et al (also in Brain, 2019) recently reported that the PM network accumulates beta-amyloid in aging, which affects performance on spatial memory tasks and also neural activity in PM regions. Since the authors seem to imply that their findings could be developed as a biomarker, it might be worthwhile considering how this biomarker corresponds to age-related pathology.

Reviewer #2 (Remarks to the Author):

"Aging alters neural event segmentation in the hippocampus and posterior-medial network"

The authors proposed to investigate age-related changes in event segmentation by examining the association between age and brain activity at event boundaries while viewing a short movie. The authors reported that age was associated with a decrease in activity in the posterior hippocampus and a set of posterior medial cortical regions, and an increase in activity in the medial prefrontal cortex and middle temporal gyrus. They also reported that boundary-evoked activity in the posterior hippocampus was associated with a separate measure of episodic memory (above and beyond the effect of age). Together, these results suggest a relationship between age-related changes in event segmentation and episodic memory.

The study examines an important topic that warrants further investigation and takes advantage of the large sample size in the CamCAN dataset. However, a few major concerns need to be addressed before one can fully understand the results.

Major concerns:

1. Throughout the manuscript, the authors report significant activation for boundaries and a correlation of age with activity at the boundaries. However, there are two critical aspects of the analysis that are needed to understand whether the reported results have anything to do with boundaries, per se, or just movie viewing in general. The authors compare boundaries during movie viewing to a control condition that contains 'visual change', but while necessary is not sufficient to support the specific conclusions about event boundaries made in the study. In other words, it is possible that the activation reported or the correlation stems from processes that are simply related to movie viewing and are not captured in the visual change regressors, but may nevertheless not be specific to boundaries. The authors can examine this possibility by including an additional regressor in the fMRI analysis for non-boundary event time points. This regressor would potentially account for activation related to cognitive processes that are involved in viewing the movie, but are not related to

the processing of boundaries. However, this would then allow a critical contrast of boundary to non-boundary movie viewing to see if indeed there are ageing-related changes in event boundary activation or more broadly during movie viewing.

2. A second concern is that while the authors do report a control of perceptual change, they report significant activation for boundary (probably compared to baseline?) and a significant activation for visual change (again, compared to baseline?). However, while this is reported for the whole brain analysis, it is not for the primary ROI analysis. Thus, for the primary ROI analysis, there is no control for boundary activation. Second, for either ROI analysis or whole-brain analysis, the appropriate procedure would be to report the contrast comparing the two conditions. For the correlation, as in (a), the authors would need to show that the correlation for boundary is significantly higher compared to visual change.

3. The boundaries in the study were identified based on a separate group of participants. However, the authors mention that segmentation in older adults is less consistent across participants. Thus, the usage of another group of raters, rather than the viewers themselves is, unfortunately, problematic. It may be that lower activity reported at boundaries may be simply because the time points that were used were not consistently perceived as boundaries in the older adults, such that the authors mix boundary and non-boundary activation. The group included 16 observers (likely young? This was not clear in my read of the original Ben Yaakov and Henson study, though even if age was matched, it may not solve the issue since the problem is inconsistency even across older adults), the threshold for boundary was 50%, meaning that it might that be as little as 8 observers indicated a time point is a boundary. Further, boundaries were depicted as agreement within 5s of the group mean, raising the concern that older adults inconsistently perceive the boundary later or earlier than the chosen time frame. The authors do show that the difference between age groups is not due to the peak activation being shifted at older adults, and that is a good check. However, inconsistency among adults in the timing of boundary detection, or in detection itself, might result in lower peak on average, as observed.

4. The authors report that boundary-activity is not correlated with other neuropsychological measures than the Logical Memory test performance. However, to argue specificity, as the authors do (l. 171), it is critical to also report whether the correlation with Logical Memory test performance holds when controlling for other neuropsychological measures. Particularly, the authors should control for other neuropsychological factors similar to how age, other ROIs, and head motion were controlled for.

Minor concerns:

1. It would be informative to see a reporting of the stepwise regression in more detail. Specifically, how does the model decide which variables to exclude/include, and what variables were eventually included in the reported model. Further, the authors should report the actual statistics on the factors that were included in the step-wise regression. Since p in some was $> .05$, it will be valuable to know the correlation value and the specific statistics, to evaluate specificity, or to consider possible mediation.

2. For the analysis of PM vs. AT regions – is the correlation in pHPC/PM higher than that in the control regions?

3. Related to major point (3): It would be interesting to investigate which cognitive factors that decline with age are associated with the decrease in boundary-evoked activity. To that end, the authors can run a multiple regression/step-wise regression with boundary-activity as the dependent variable, and the neuropsychological factors as predictors.

4. Regarding the correlation between pHPC and ANG:

a. This analysis should be compared to the other control conditions as well (visual change and non-boundary, as suggested above).

b. The authors should report exactly what measures they correlated between pHPC and ANG – did

they do a PPI? Correlated average activity across participants? Some type of "trial"-evoked activity?

5. l. 97: Did the authors define anterior and posterior hippocampus based on connectivity or on the uncus apex? This should be stated more clearly in the results.

6. l. 462-3: What do the authors mean by "set" of betas? Based on the Methods, there seems to be only one beta per condition.

7. l. 140: The authors refer to the memory test as both recall and recognition. This is confusing – was the test a recall test or a recognition test?

8. l. 501-502: Please clarify: are the stories in the logical memory test different than the ones the participants view? If so, as mentioned in another section of the manuscript, this is unclear here.

Reviewer #3 (Remarks to the Author):

In this manuscript, Reagh&Ranganath address an important and timely question, asking whether boundary-evoked activity in the hippocampus and PM network is affected by aging and whether these changes are linked to episodic memory performance. They analyze a very large dataset of 546 participants across a large age range and find strong correlations between boundary-evoked activity and age in the posterior hippocampus and several posterior-medial network regions. While these results are certainly interesting, I have a few concerns that detract from the potential impact of the paper, primarily a critical confounding factor (comment 1b) that is not currently addressed.

Main concerns:

1. The decreased boundary response in pHPC with age is interpreted as evidence of hippocampal involvement in segmentation and indication that changes in boundary perception may be driven by this region. However, as the authors mention, perception of boundaries tends to differ between younger and older adults, and in particular is less consistent among older individuals.

a. First, the age range of the independent observers should roughly match the age range of the fMRI participants. Otherwise, any effect of age could be due to differences in the similarity of the participants' boundaries to those of the observers.

b. However, even if the observers are age-matched, the main finding regarding older adults is of reduced consistency in their segmentation. This means there is a very plausible alternative interpretation of the results – that the older adults are more varied in where their boundaries are, leading to reduced hippocampal activity at group-defined boundaries. If the participants themselves had segmented the movie, there may be no decline with age in the same dataset. While (a) can be relatively easily addressed by collecting segmentations from older observers, I do not see how the second confound can be accounted for in the current dataset, and it has significant bearing on the ability to interpret the main finding (as well as the correlation with memory). I am happy to be proven wrong on this and would welcome the authors' thoughts.

2. When correlating the pHPC and ANG in different age groups, there is a lower correlation for older adults. This could arise from the lower response in these regions, as a lower response is linked to lower snr and can lead directly to lower correlations. It is difficult to interpret this finding without accounting for the amplitude differences between the groups.

Minor concerns:

1. The Introduction should explicitly state that increased hippocampal activity at boundaries has already been identified on the same dataset. I am sure this is accidental, but the current phrasing is somewhat misleading, as this is described as one of the predictions of the study (lines 75-77), despite having already been demonstrated. Similarly, the authors may wish to mention in the section about

boundaries and visual flow that HPC boundary-evoked responses have already been shown not to be driven by low-level visual changes in this dataset, although with a different method of assessing perceptual change.

2. There seem to be at least two places where the references may have been confused:

a. In the Introduction, lines 45-48, is Chen et al. indeed the intended citation? The Chen 2017 study finds that hippocampal inter-subject correlation during scenes is linked to subsequent recall rather than boundary-evoked activity.

b. In line 425, are the citations in the sentence the correct ones?

3. In what sense did activity at boundaries in the pHPC 'uniquely predict' recognition memory (line 140)? Does this mean of all ROIs in the stepwise regression it was the only one to predict story recall? Did any of the cortical PM network regions also predict memory?

4. While the pHPC remains significant in explaining delayed recall memory after accounting for age, it is just-significant (especially as both immediate and delayed recall were tested in both pHPC and aHPC). While this is stated in the results, perhaps the description of this finding in the discussion could be toned down a bit. I would mention in the discussion as well that age was by far the primary explanatory variable, and that the relationship between pHPC and delayed recall was driven primarily by age, but remained significant. To further probe this finding, it may be interesting to test whether pHPC activity predicts delayed recall within each of the age groups.

5. In line 161, the authors mention there was no significant relationship of pHPC boundary-evoked activity with other neuropsychological tests. As evidence they mention all p-values are larger than 0.05. It would be preferable to give the minimal p-value of these tests, so the reader can assess whether some tests were related to pHPC activity but did not reach significance, or whether there was clearly no relationship.

6. Could the authors provide some rationale for the way in which they defined the boundaries?

7. I found the edge pixel analysis a bit unclear. How were the edge pixels calculated? Why was this method chosen to assess visual flow? If I understand correctly, the calculation was the proportion of edge pixels in each frame (averaged within TRs) – wouldn't this predict high-frequency visual information within each time point, rather than visual change across time points?

8. In lines 482-484, adding the range of ages, in addition to the mean, would be informative.

9. The authors mention in lines 383-384 that all participants reported not having seen the movie. It would be more accurate to say 'most participants', as some of the participants did report having seen the movie.

Reviewer #1 (Remarks to the Author):

This manuscript reports on a study of aging in which 546 participants over a large age range watched a short film while their brain activity was recorded using fMRI. The variable of interest was regional activity at the time of event boundaries (defined using a separate group of subjects). The authors report increased hippocampal activity at the time of these event boundaries, but only the posterior hippocampus showed a reduction in this activation as a function of age. This reduction was related to memory for stories on the WMS logical memory subtest. Boundary event-related activation increases were also seen in components of the posteromedial (PM) network, specifically angular gyrus, posteromedial cortex, and parahippocampal cortex and these also showed reduced activation with advancing age. Medial PFC and medial temporal gyrus showed similar activation but increases with age. There was no boundary evoked activity in the anterior temporal (AT) network and consequently no relationship between activation and age. These ROI-based findings were confirmed using voxelwise approaches and, finally, correlations between angular gyrus and hippocampus were seen in younger individuals but these also declined with age.

Overall, I found this manuscript very enjoyable and even exciting to read. The writing is clear and the manuscript is illustrated nicely. The authors tackle a very interesting question, and they bring together both cognitive and neuroscience approaches to answer questions of translational neuroscience interest. The findings are undoubtedly novel and would be of interest to many people working in the fields of both cognitive aging and memory. I have no substantive concerns about methodological factors.

We thank the Reviewer for this constructive feedback regarding the presentation and significance of these results.

I do have a few questions that arose while reading the manuscript and I think the authors might want to consider these points which I am listing in order of importance:

1. I realize that this is one of several reports implicating posterior brain regions, parts of the PM network, in activation during event boundaries. However, I wonder how much this association is driven by the nature of the stimuli as much as by the “boundary detection” aspect of the task. The PM network shows functional specialization for spatial information, and boundary events, particularly in films (which is the main sort of stimulus

used in these studies), are inherently spatial – different events in films are, after all, usually demarcated as different “scenes”. I realize that these two concepts are so confounded that they may be impossible to differentiate experimentally (even written or auditory stimuli are likely to induce imagery). And I am definitely not asking the authors to perform another experiment to resolve this. And the authors addressed the perceptual aspects of the task, so it should be clear that is not what I’m talking about. However, at the very least it would be interesting to hear the authors’ views on whether the localization of these findings to PM reflect the event-detection aspect of the task, or the spatial aspect of the task, or whether this is impossible to resolve. And as a related question, do the authors have any measure of how often a boundary event coincided with a scene change (as opposed, perhaps to an “action change”)?

This is an excellent point. We agree with the Reviewer that many instances of event boundaries in films (which are often choreographed and shot to emphasize transitions across scenes) are triggered by spatial context changes. Dr. Aya Ben-Yakov, who obtained the boundary ratings used in our analyses provided several additional metrics associated with these boundaries. One of these is an indication of whether each boundary is associated with a major shift in spatial context (judged by research assistants viewing the video clip). As might be suspected, the majority of these boundaries involve such a shift in spatial context – 9 out of 12. A greater number of non-spatially-driven event boundaries (driven by dialogue or tonal shifts, for instance) exist in the original film, but unfortunately there are too few of these nonspatially-driven event boundaries in the 8 minute version of the film used in the CamCAN study. Accordingly, we did not have enough observations to separately model and contrast spatially-driven and non-spatial boundaries in our univariate fMRI model in the present dataset. We now briefly discuss this issue and the implications it may carry in the Discussion:

The paradigm used in the CAM-CAN dataset provides limited insight into the precise factors that triggered activation at event boundaries. Our analyses suggest that low-level visual information does not account for PM network activity at event boundaries, or changes with age. In terms of higher-order factors, it is known that spatial context changes reliably trigger event segmentation (Magliano & Zacks, 2011), and many (but not all) of the boundaries modeled in this 8-minute dataset involved a significant change in spatial context (9 out of 12). However, it is well established that event segmentation can also be triggered by prediction errors even in the absence of significant changes in spatial context or sensory input (Zacks et al., 2010). Although we do not have enough event boundaries in the present data to separately examine activity during spatial context changes and activity during nonspatial changes or prediction errors, this distinction can be fruitfully explored in future studies.

2. I found the observations of age-related boundary activity in posterior components of the PM network to be interesting, and also consistent with other reports. One new finding is the relationship between posterior hippocampus and memory performance on a narrative task unrelated to the fMRI experiment. Did the authors test for this association in angular gyrus, posteromedial cortex, and parahippocampal cortex? And

is there any relationship between the strength of the relationship between posterior hippocampal signal change and angular gyrus signal change and whether angular gyrus is related to memory performance? In other words, I would expect that if the PM regions are highly correlated to posterior hippocampus, they too would be related to memory function, while if this correlation is lost, they might not be.

Thank you for this suggestion. We did indeed examine whether other regions of the PM network were correlated with memory performance. However, we did not observe significant relationships between behavioral variables and activity of any other region besides posterior hippocampus (though angular gyrus did appear to show a marginal relationship, qualitatively in the expected direction, for delayed recall: $r = 0.083$, $p = 0.052$). This may reflect the fact that posterior hippocampus showed the strongest boundary-evoked activation observed in our analyses, and the fact that it showed the strongest relationship with age. In the revised manuscript, we have added the correlations between story memory performance and activity in PM network ROIs in a new Table 2, seen below:

Variable	Coefficient	Std. Error	t	p
Constant	0.3505	1.048	0.334	0.738
pHPC	3.0788	1.455	2.116	0.035*
aHPC	1.9977	1.782	1.121	0.263
PMC	-0.8259	1.808	-0.457	0.648
ANG	0.6639	2.124	0.313	0.755
PHC	0.6288	2.637	0.241	0.809
MTG	0.4054	1.494	0.271	0.786
mPFC	-0.6881	2.175	-0.316	0.752
EVC	-2.3315	1.430	-1.630	0.104
PRC	-0.0061	1.677	-0.004	0.997
TP	0.0957	0.808	0.118	0.906
AMY	-0.9888	1.583	-0.625	0.532
Motion	1.5441	1.578	0.978	0.328
Verbal Fluency	-0.0107	0.045	-0.237	0.812
Visuospatial	0.0302	0.041	0.744	0.457

Word Memory	0.0033	0.026	0.126	0.900
Log Mem Imm	0.8831	0.027	33.250	<0.001*
Age	-0.0194	0.007	-2.905	0.004*

3. I think the voxelwise analysis makes the participation of the AT network in boundary event detection unlikely, and I liked the use of AT ROIs as controls. But these AT ROIs were generally small and at the edges of brain, so measurement could be a bit difficult. Could the authors test larger ROIs, particularly amygdala, in their suite of ROIs shown in figure 6?

Thank you for this suggestion. Our AT network ROIs are, on average, a bit smaller than the PM network ROIs. Following the Reviewer's suggestion, we added the amygdala to the suite of ROIs. We found that the amygdala *did* in fact show significant boundary-evoked activity across our sample. However, we did not observe a relationship between activation and age in the amygdala, keeping with the finding of selective age-related changes in PM network regions. Please see revised Figures 4 and 6 below, as well as our revised Results section incorporating this findings:

Figure 4: Regions of Interest (ROIs). ROIs are displayed as surface maps normalized to MNI space. EVC = early visual cortex, ANG = angular gyrus, MTG = middle temporal gyrus, PMC = posterior-medial cortex, HPC = hippocampus (anterior/posterior subdivisions not displayed here), PHC = parahippocampal cortex, PRC = perirhinal cortex, AMY = amygdala, TP = temporal pole, mPFC = medial prefrontal cortex.

Figure 6: Anterior-temporal regions and early visual cortex do not show age-related changes. Boundary-evoked responses were not observed in (A,B) perirhinal cortex or (C,D) temporal poles. Boundary-evoked responses were observed in (E,F) early visual cortex and (G,H) amygdala, but these did not significantly change with age.

Results, Page 15:

We additionally found boundary-evoked responses in the amygdala (AMY; ($t_{(545)} = 5.133$, $p = 3.97e^{-07}$) (Fig. 6F). However, critically, we did not observe a relationship between AMY activity and age ($r = -0.026$, $p = 0.548$) (Fig. 6E). Thus, the age-related differences in activity driven by event boundaries in the PM network are not seen in the AT network, or in early visual processing regions.

We finally included an early visual cortex (EVC) ROI as a control region, testing whether age-related changes in boundary evoked activity might have been driven by low-level perceptual processes. We did observe significant boundary-evoked activity in EVC ($t_{(545)} = 4.177$, $p = 3.45e^{-05}$) (Fig 6H), which is perhaps to be expected, as event boundaries in naturalistic stimuli are often associated with visuospatial changes [1-3,17]. However, we did not observe a significant relationship between EVC activity and age ($r = -0.057$, $p = 0.177$) (Fig. 6G).

4. I'm not clear how the authors assessed significance in figure 8. The proper way is an interaction. Did they test a correlation x age interaction?

Based on feedback from other Reviewers, we ultimately decided to remove the pHPC-ANG correlation from the paper. Though we think that this is an interesting result, it is one that is not central to the focus of the paper and main analyses. In case the Reviewer is still curious, here is more information on how that analysis was conducted:

As we report in the Results, we find an overall significant correlation between these two regions, but we wanted to know whether the correlation we observed was stronger in some age groups than others (i.e., whether younger adults showed a stronger correlation than middle-aged adults and older adults). To do this, we used Fisher z-transform over the Pearson r coefficients such that the correlation values are normally distributed and, thus, can be compared statistically. Each Pearson r was converted into a z-score, and those z-scores were subtracted and assessed for significance. For example, the z-score of the correlation between pHPC and ANG was derived for younger adults and for older adults, the latter z-score was subtracted from the former, and the difference was divided by the standard error of the z-distribution (see Steiger, *Psychological Bulletin*, 1980). This z-test provides a means of conducting pairwise comparisons among related correlation values. We hope this clarifies the matter. We are unfortunately a bit confused about how to analyze these data to test for an interaction, as age is really the only factor present in the analysis. The correlation between the two regions is the dependent variable rather than an independent factor, so it is not clear how we could go about assessing a correlation x age interaction.

5. There are a number of papers in the amyloid and tau literature related to “pathological aging” that the authors might wish to read and include. Jones et al (Brain 2015) defined a course of network failure beginning posteriorly and spreading anteriorly, providing a more meaningful explanation of the “posterior-anterior” shift. Maass et al (also in Brain, 2019) recently reported that the PM network accumulates beta-amyloid in aging, which affects performance on spatial memory tasks and also neural activity in PM regions. Since the authors seem to imply that their findings could be developed as a biomarker, it might be worthwhile considering how this biomarker corresponds to age-related pathology.

Thank you, we agree that these papers are highly relevant and merit discussion in light of some of the posterior vs. anterior effects we observed. We have incorporated this important work into our review of relevant aging-related phenomena, and have additionally fleshed out the potential of measures such as this as biomarkers in the Discussion:

A recent study by Jones et al. [51] suggests that dysfunctions in functional activity in the more posterior aspect of the PM Network occur early in disease progression, and later ‘cascade’ into more anterior brain networks. This distinction could explain the dissociation between posterior and more anterior PM Network regions in the present data. Critically, the PM Network dysfunction reported by Jones et al. also precedes amyloid accumulation, suggesting that regional dysfunction may be among the earliest possible biomarkers. PM Network functions such as perceiving and remembering lifelike scenes may thus be selectively vulnerable in certain aging individuals, particularly so with amyloid accumulation, as is reported by Maass and colleagues [52]. We suggest that behavioral and neural measures related to complex, naturalistic events may reflect a process that taps into PM network integrity, and our results show that

they are sensitive to aging. Importantly, these functional measures relate to processing of lifelike situations, and thus may reflect cognitive status in a way that mirrors daily functioning more closely than the majority of neuropsychological tests. Accordingly, studies of event cognition and event representation may be a promising direction for characterizing preclinical Alzheimer's Disease.

Reviewer #2 (Remarks to the Author):

“Aging alters neural event segmentation in the hippocampus and posterior-medial network”

The authors proposed to investigate age-related changes in event segmentation by examining the association between age and brain activity at event boundaries while viewing a short movie. The authors reported that age was associated with a decrease in activity in the posterior hippocampus and a set of posterior medial cortical regions, and an increase in activity in the medial prefrontal cortex and middle temporal gyrus. They also reported that boundary-evoked activity in the posterior hippocampus was associated with a separate measure of episodic memory (above and beyond the effect of age). Together, these results suggest a relationship between age-related changes in event segmentation and episodic memory.

The study examines an important topic that warrants further investigation and takes advantage of the large sample size in the CamCAN dataset. However, a few major concerns need to be addressed before one can fully understand the results.

Major concerns:

1. Throughout the manuscript, the authors report significant activation for boundaries and a correlation of age with activity at the boundaries. However, there are two critical aspects of the analysis that are needed to understand whether the reported results have anything to do with boundaries, per se, or just movie viewing in general. The authors compare boundaries during movie viewing to a control condition that contains ‘visual change’, but while necessary is not sufficient to support the specific conclusions about event boundaries made in the study. In other words, it is possible that the activation reported or the correlation stems from processes that are simply related to movie viewing and are not captured in the visual change regressors, but may nevertheless not be specific to boundaries. The authors can examine this possibility by including an additional regressor in the fMRI analysis for non-boundary event time points. This regressor would potentially account for activation related to cognitive processes that are involved in viewing the movie, but are not related to the processing of boundaries. However, this would then allow a critical contrast of boundary to non-boundary movie viewing to see if indeed there are ageing-related changes in event boundary activation or more broadly during movie viewing.

Thank you, this is an important point to consider. We did indeed include a regressor of 12 within-event timepoints as the Reviewer suggests. Please see our revised Figure 2

below showing activation maps related to this regressor, which can be easily compared to activity related to event boundaries. To be clear, our contrast is event boundaries, or within-event timepoints compared to baseline, which includes nuisance regressors for motion and drift as well as a continuous regressor for high-frequency visual information. In general, it seems clear that the boundary and non-boundary timepoints do not elicit overlapping activity throughout most regions in the brain. Moreover, for our main analyses over ROIs, we now subtract within-event activity from boundary-evoked activity. The pattern of results remained solid even with these changes, bolstering the argument that aging affects segmentation-related brain responses.

Additionally, please see our revised section of the Results describing the voxelwise analysis:

2.4. Voxelwise analysis: boundary-evoked activity in the hippocampus and Posterior Medial network is distinct from within-event timepoints.

We next ran a whole-brain voxelwise analysis to assess the selectivity of boundary-evoked responses to the PM network. Consistent with results from the ROI analysis, the exploratory voxel-based analysis revealed that boundary-evoked activity was centered on the PM network (voxelwise $q < 0.05$) (Fig. 7,

blue). This aligns with our ROI analyses, and is consistent with prior reports indicating BOLD activity in PM regions is sensitive to naturalistic event structure [6,10,11,28]. We additionally modeled within-event timepoints to assess regional specificity of boundary-evoked responses versus within-event activity. As can be seen in Figure 7 (red), within-event timepoints feature very little overlap in terms of regional activity compared to event boundaries. Taken together, this confirms that our hippocampal and PM network ROIs are reliably driven by event boundaries, and that activity in these regions is largely separate from activity observed within an event.

Moreover, though we have not included this as a figure in the paper, we conducted an exploratory analysis over a cluster of significant voxels at the group level, and looked for correlations with age in this cluster (though this is, of course, circular ROI selection, this is only meant to be a confirmatory analysis). In a cluster of voxels in the dorsomedial PFC (the lower-left red cluster in Figure 2 above), we indeed find significant within-event (minus boundary-evoked) activity, but no significant change with age:

Finally, please see our revised section of the Methods relevant to this new approach:

The GLM included a discrete regressor for human-labeled event boundaries, defined by a sample of 16 independent observers gathered by Ben Yakov and Henson [11]. Briefly, these event boundaries were gathered by participants who, when watching the 8-minute video stimulus, were simply instructed to press a key when one meaningful unit (i.e., an event) ended and another began, in line with prior studies [19-22]. We included boundary timepoints for which at least half of the participants from the sample agreed (within 5s of a group-means boundary time), and which were no closer than 6s to one another, totaling 12 event boundaries across the time series. The logic for this selection process was to include timepoints that we were confident featured what the majority of

observers would perceive as an event boundary. We attempted to control for low-level visual information in our GLM by modeling edge pixels in the video, and entering the proportion of edge pixels in each time point as a continuous regressor. Edge pixels were calculated via a python routine which read in the video stimulus, split it into its constituent frames, and in an automated fashion performed edge-detection on each frame (using python package opencv). The proportion of edge pixels to total pixel count was calculated (NumPy) for each frame, and was output into a comma-separated value file. We then resampled frame-by-frame edge information to correspond to the temporal scale of the fMRI data by averaging across adjacent frames within the interval of each TR (2470ms). This resultant temporally-smoothed vector served as our estimate of low-level visual information in each timepoint of the video. This approximates an envelope of framewise high-frequency visual information in the video that is modeled independently of event boundaries. This high-frequency visual information vector was entered into the GLM as a regressor of non-interest. Twelve continuous nuisance regressors were also included to account for motion (6 motion regressors – x, y, z, pitch, roll, yaw – plus the derivative of each). These thirteen nuisance regressors served as the model baseline. We also included a nuisance regressor to account for linear drift. Finally, we included two regressors of interest: event boundary timepoints, and within-event timepoints to be contrasted with event boundaries. The within-event regressor consisting of 12 timepoints determined by calculating the average elapsed time between event boundaries, and distributing them evenly throughout the video (ensuring that they fell within an event, no more than 6sec from an event boundary). The resultant GLM produced two beta images of interest: (1) one corresponding neural responses at event boundaries, but not within-event, and (2) one corresponding to neural signals that were present during within-event timepoints, but not event boundaries. Our hypotheses were specifically about the former condition. Thus, to control for within-event activity, we subtracted within-event betas from boundary-evoked betas.

2. A second concern is that while the authors do report a control of perceptual change, they report significant activation for boundary (probably compared to baseline?) and a significant activation for visual change (again, compared to baseline?). However, while this is reported for the whole brain analysis, it is not for the primary ROI analysis. Thus, for the primary ROI analysis, there is no control for boundary activation. Second, for either ROI analysis or whole-brain analysis, the appropriate procedure would be to report the contrast comparing the two conditions. For the correlation, as in (a), the authors would need to show that the correlation for boundary is significantly higher compared to visual change.

We apologize for the lack of clarity on this point. In our univariate regression model, our baseline condition is essentially a 'null' model containing terms for linear drift and twelve motion-related regressors, and it is against this condition that the betas (converted to percent signal change) for event boundary and within-event timepoints are derived. We note that, in our revised analyses, we have modeled the high-frequency visual

information as a nuisance regressor of non-interest (e.g., alongside motion) in order to capture boundary-evoked activation that cannot be accounted for by low level visual change. This also allowed for a cleaner comparison of boundary-evoked activity compared to non-boundary activity, as in our revised analyses and figures. That is, for all of our main correlation analyses and averaged activity across age groups, the dependent variable is now percent signal change of boundary-evoked activity *minus* within-event activity. Thus, the correlations and across-group assessments now account for corresponding within-boundary activity. Please see our revised portion of our Methods in response to your above comment to see our description of this approach in the manuscript.

3. The boundaries in the study were identified based on a separate group of participants. However, the authors mention that segmentation in older adults is less consistent across participants. Thus, the usage of another group of raters, rather than the viewers themselves is, unfortunately, problematic. It may be that lower activity reported at boundaries may be simply because the time points that were used were not consistently perceived as boundaries in the older adults, such that the authors mix boundary and non-boundary activation. The group included 16 observers (likely young? This was not clear in my read of the original Ben Yaakov and Henson study, though even if age was matched, it may not solve the issue since the problem is inconsistency even across older adults), the threshold for boundary was 50%, meaning that it might that be as little as 8 observers indicated a time point is a boundary. Further, boundaries were depicted as agreement within 5s of the group mean, raising the concern that older adults inconsistently perceive the boundary later or earlier than the chosen time frame. The authors do show that the difference between age groups is not due to the peak activation being shifted at older adults, and that is a good check. However, inconsistency among adults in the timing of boundary detection, or in detection itself, might result in lower peak on average, as observed.

Thank you, this is a very good point. Ultimately, as the Reviewer suggests, there is no real way of ensuring that older participants were not simply 'missing' the relevant event boundaries in these data on the back-end. The Reviewer is correct that the original 16 observers in the Ben Yakov and Henson study were healthy young participants. As the Reviewer suggests, we did observe that, on average, activity at event boundaries did not seem to be 'shifted' in older adults relative to younger adults, but the lower peak may yet be due to greater variability in segmentation of older adults.

Although boundary ratings are not available for the participants in the CamCAN sample, we have followed up on the reviewer comments by collecting event boundary estimates in an older adult sample of a sample of 14 older adult raters (mean age = 73.83 years, SD = 6.27). We additionally ran a new sample of 14 younger adult raters (mean age = 20.15 years, SD = 2.58) to compare rating distributions (the boundary ratings provided by Ben Yakov et al. are average boundaries across participants with at least two raters agreeing, whereas we wanted a full, matched dataset to compare across age groups). Our logic here was that although these older adults comprise an obviously distinct sample from the individuals whose brain activity we are analyzing, we should

nonetheless be able to determine that, in general, older adults are identifying the 12 event boundaries we included in our model. In sum, we did not observe evidence that older adults were not segmenting events in a way that was significantly different than younger adults for this particular stimulus. When comparing the older adult raters to the 12 event boundary timepoints we used in our regressor for fMRI analyses, we observed that all of them were identified by at least half of our older adult sample (i.e., the criterion for these 12 ‘maximal agreement’ timepoints, which is actually a fairly stringent criterion amongst the few reports in the literature that have studied neural responses to event boundaries). On average, the boundaries included in fMRI analyses were identified by older adults 86% of the time across the 14 participants. Moreover, comparing older adults with the new sample of younger adults we collected demonstrates very high overlap across age groups in boundary ratings (see below). Finally, we compared segmentation agreement within older adults to agreement within younger adults, as well as young-young agreement to old-young, and old-old agreement to old-young. There were no differences between groups in segmentation agreement among older adults ($r = 0.69$) and among younger adults ($r = 0.73$) ($z = 0.19$, $p = 0.85$), or between either group and agreement across younger and older adults ($r = 0.65$) (young-young vs. young-old: $z = 0.36$, $p = 0.719$; old-old vs. young-old: $z = 0.17$, $p = 0.87$). Together, we believe this amounts to fairly compelling evidence that, despite at least some appreciable age-related variability in where boundaries are perceived, the timepoints we are modeling in this dataset are *not* likely mischaracterizing times during which boundary-evoked activity is driven in older adults. We highlight this point in an existing portion of the Results section as follows:

Finally, visualizing the data as a time-course of activity averaged across the modeled event boundaries, it is clear that the age-related decline in pHPC reflects a reduction in the amplitude of the hemodynamic response, rather than a fundamentally different shape, or a temporal shift of the response (Fig. 2A). Thus, age-related differences are not due to a failure to account for a shifted peak of the response. It is also clear that boundary-evoked response time courses and their relationship with age are less robust in aHPC than pHPC (Fig. 2B).

We additionally added an additional section to the Results to address this issue:

2.5. Older and younger participants segment the video stimulus similarly. One possible explanation for age-related changes in boundary-evoked responses is inconsistency of perceived event boundaries in older adults. Though we cannot rule out that older participants in the CamCAN dataset simply ‘missed’ a number of the 12 event boundaries we modeled, we addressed this possibility in a separate sample of participants. We collected a sample of 14 older adults to provide some evidence as to whether there were age-associated differences in boundary detection. We did not find evidence for any such differences. In our smaller sample of older adults, at least half of the sample identified each of the 12 boundaries (i.e., the criterion for the 12 ‘maximal

agreement' boundaries from the sample of young adults who provided the ratings), and the boundaries were overall identified by our sample 86% of the time (compared to 82% of the time in our younger adult sample). Moreover, there were no differences between groups in segmentation agreement among older adults ($r = 0.69$) and among younger adults ($r = 0.73$) ($z = 0.19$, $p = 0.85$), or between either group and agreement across younger and older adults ($r = 0.65$) (young-young vs. young-old: $z = 0.36$, $p = 0.719$; old-old vs. young-old: $z = 0.17$, $p = 0.87$). Thus, though it cannot be absolutely ruled out, it is unlikely that older adults simply perceived the key event boundaries differently than young adults.

Finally, please see a plot below comparing our 14 older adults with the new sample of younger adults who provided event boundary ratings. It is immediately apparent from the plot that the major event boundaries are being consistently identified by a majority of both young and older participants:

4. The authors report that boundary-activity is not correlated with other neuropsychological measures than the Logical Memory test performance. However, to argue specificity, as the authors do (l. 171), it is critical to also report whether the correlation with Logical Memory test performance holds when controlling for other neuropsychological measures. Particularly, the authors should control for other neuropsychological factors similar to how age, other ROIs, and head motion were controlled for.

Thank you, this is a great point. In the revised manuscript, we conducted a least squares multiple linear regression with *all* neuropsychological measures of interest in the model. Our results here are virtually unchanged: for predicting logical memory delay

scores, age ($t = -2.959$, $p = 0.003$) and logical memory immediate ($t = 33.251$, $p < 0.001$) predict the majority of the variance, but posterior hippocampal activity at event boundaries ($t = 3.107$, $p = 0.034$) is still the only other measure – among all other regions or neuropsychological tests – which also predicts significant variance in delayed logical memory recall. Please see our revised Methods section on this for an updated description of the analysis:

4.8. Multiple linear regression analysis

To simultaneously assess the predictive power of all predictor variables of interest in explaining variance in story memory, we fit multiple linear regression models (linear_model.LinearRegression in scikit-learn’s linear_model package) using an ordinary least squares approach. Briefly, this approach allows us to assess the relationship between variables of interest (e.g., pHPC activity and Logical Memory delayed recall) while also accounting for other important variables of interest (e.g., age or head motion). The model predicted Logical Memory performance immediately and at a delay, including all ROIs and neuropsychological test scores, as well as age and head motion (average framewise displacement) as predictor variables.

Minor concerns:

1. It would be informative to see a reporting of the stepwise regression in more detail. Specifically, how does the model decide which variables to exclude/include, and what variables were eventually included in the reported model. Further, the authors should report the actual statistics on the factors that were included in the step-wise regression. Since p in some was $> .05$, it will be valuable to know the correlation value and the specific statistics, to evaluate specificity, or to consider possible mediation.

Thank you for pointing out the ambiguity in this analysis. As noted above, we have now moved toward a simpler OLS multiple linear regression model. We now include *every* factor into the analysis to be evaluated simultaneously. The variance explained by each regressor can be seen in the following table (now Table 2 in the manuscript), showing that the significant predictive value of pHPC on Logical Memory delayed recall is actually quite specific among the set of predictor variables (in fact, the only other significant predictors are Logical Memory Immediate, and Age):

Variable	Coefficient	Std. Error	t	p
Constant	0.3505	1.048	0.334	0.738
pHPC	3.0788	1.455	2.116	0.035*
aHPC	1.9977	1.782	1.121	0.263
PMC	-0.8259	1.808	-0.457	0.648

ANG	0.6639	2.124	0.313	0.755
PHC	0.6288	2.637	0.241	0.809
MTG	0.4054	1.494	0.271	0.786
mPFC	-0.6881	2.175	-0.316	0.752
EVC	-2.3315	1.430	-1.630	0.104
PRC	-0.0061	1.677	-0.004	0.997
TP	0.0957	0.808	0.118	0.906
AMY	-0.9888	1.583	-0.625	0.532
Motion	1.5441	1.578	0.978	0.328
Verbal Fluency	-0.0107	0.045	-0.237	0.812
Visuospatial	0.0302	0.041	0.744	0.457
Word Memory	0.0033	0.026	0.126	0.900
Log Mem Imm	0.8831	0.027	33.250	<0.001*
Age	-0.0194	0.007	-2.905	0.004*

2. For the analysis of PM vs. AT regions – is the correlation in pHPC/PM higher than that in the control regions?

We first want to note that, in the revised manuscript, we have omitted the pHPC-ANG correlational analysis as it is not central to the questions we sought to answer with this study. However, to address the Reviewer's question: on average, we do in fact see significant correlations between pHPC and all PM regions ($r = 0.1056$, $p = 0.014$), but not between pHPC and all AT regions ($r = 0.015$, $p = 0.734$). However, (1) we note that the pHPC-PM correlation is largely driven by ANG, and (2) using a Fisher z-transformed comparison of the two averaged correlations across PM and AT networks, the direct comparison is not significant ($z = 1.51$, $p = 0.131$).

3. Related to major point (3): It would be interesting to investigate which cognitive factors that decline with age are associated with the decrease in boundary-evoked activity. To that end, the authors can run a multiple regression/step-wise regression with boundary-activity as the dependent variable, and the neuropsychological factors as predictors.

Thank you for this suggestion. We ran this analysis, predicting pHPC activity. Results can be seen in the image below. In general, Age and Logical Memory Delayed Recall significantly predict variance in pHPC boundary responses, with the only other predictive variable being ANG activity (tracking with our correlational analyses over these two regions). However, upon closely reviewing the data and manuscript in light of this and other feedback, we ultimately have decided to remove the pHPC-ANG correlation from the paper, as mentioned in the response to the prior point. Though we think that this is an interesting result, it is one that is not central to the main analyses, and does not lend itself to clear interpretation in light of our other findings. Please see the results of this regression analysis in the image below:

OLS Regression Results						
=====						
Dep. Variable:	pHPC		R-squared:	0.146		
Model:	OLS		Adj. R-squared:	0.118		
Method:	Least Squares		F-statistic:	5.307		
Date:	Tue, 10 Dec 2019		Prob (F-statistic):	6.12e-11		
Time:	13:50:59		Log-Likelihood:	703.33		
No. Observations:	546		AIC:	-1371.		
Df Residuals:	528		BIC:	-1293.		
Df Model:	17					
Covariance Type:	nonrobust					
=====						
	coef	std err	t	P> t	[0.025	0.975]

const	0.1268	0.031	4.127	0.000	0.066	0.187
aHPC	0.0137	0.053	0.258	0.796	-0.091	0.118
PMC	0.0039	0.054	0.073	0.942	-0.102	0.110
ANG	0.1394	0.063	2.214	0.027	0.016	0.263
PHC	-0.0243	0.079	-0.309	0.757	-0.179	0.130
MTG	0.0282	0.044	0.634	0.526	-0.059	0.116
mPFC	-0.0264	0.065	-0.408	0.684	-0.154	0.101
EVC	0.0059	0.043	0.138	0.890	-0.078	0.090
PRC	0.0323	0.050	0.646	0.518	-0.066	0.130
TP	0.0030	0.024	0.125	0.901	-0.044	0.050
AMY	0.0221	0.047	0.469	0.640	-0.071	0.115
Motion	-0.0273	0.047	-0.580	0.562	-0.120	0.065
Fluency	-0.0001	0.001	-0.099	0.921	-0.003	0.003
Visuospatial	-0.0011	0.001	-0.924	0.356	-0.003	0.001
Word_Mem	0.0011	0.001	1.440	0.150	-0.000	0.003
Log_Mem_Imm	-0.0013	0.001	-0.956	0.339	-0.004	0.001
Log_Mem_Del	0.0027	0.001	2.116	0.035	0.000	0.005
Age	-0.0011	0.000	-5.464	0.000	-0.001	-0.001
=====						
Omnibus:	2.139		Durbin-Watson:	1.883		
Prob(Omnibus):	0.343		Jarque-Bera (JB):	2.106		
Skew:	0.152		Prob(JB):	0.349		
Kurtosis:	2.984		Cond. No.	1.80e+03		
=====						

4. Regarding the correlation between pHPC and ANG:
 - a. This analysis should be compared to the other control conditions as well (visual change and non-boundary, as suggested above).
 - b. The authors should report exactly what measures they correlated between pHPC and

ANG – did they do a PPI? Correlated average activity across participants? Some type of “trial”-evoked activity?

Thank you for these suggestions, which are very good ones, and for these requests for clarification. Per our response above, however, please note that we have now omitted this analysis from the manuscript. To respond to the clarification questions, these correlations depicted boundary-evoked activity in each region across participants (i.e., this amounted to asking whether, on a participant-by-participant basis, boundary-evoked activity in pHPC tracked with activity in ANG).

5. I. 97: Did the authors define anterior and posterior hippocampus based on connectivity or on the uncus apex? This should be stated more clearly in the results.

The Reviewer is correct in the latter suggestion, that we indeed divided anterior and posterior hippocampus at the uncus apex. We have clarified this in the Results as follows on Page 6:

Based on their differential connectivity with the PM and ATs networks, we defined separate regions of interest (ROIs) for the posterior and anterior (aHPC) hippocampus (divided longitudinally at the uncus apex).

6. I. 462-3: What do the authors mean by “set” of betas? Based on the Methods, there seems to be only one beta per condition.

Apologies for the typo. The Reviewer is correct, and the word “set” has been replaced.

7. I. 140: The authors refer to the memory test as both recall and recognition. This is confusing – was the test a recall test or a recognition test?

This is another typo, thank you for pointing this out. We did in fact mean to use “recall” here and not “recognition”. This has been corrected.

8. I. 501-502: Please clarify: are the stories in the logical memory test different than the ones the participants view? If so, as mentioned in another section of the manuscript, this is unclear here.

Apologies for the ambiguity. Please see our clarification in the revised Results section:

Although memory for the video stimulus used during the scan session was not assessed in this experiment, the participants completed a separate test of memory for narratives– the Logical Memory portion of the Wechsler Memory Scale. These stories, like the film used in the fMRI study, consist of a series of thematically-linked events.

Reviewer #3 (Remarks to the Author):

In this manuscript, Reagh&Ranganath address an important and timely question, asking whether boundary-evoked activity in the hippocampus and PM network is affected by aging and whether these changes are linked to episodic memory performance. They analyze a very large dataset of 546 participants across a large age range and find strong correlations between boundary-evoked activity and age in the posterior hippocampus and several posterior-medial network regions. While these results are certainly interesting, I have a few concerns that detract from the potential impact of the paper, primarily a critical confounding factor (comment 1b) that is not currently addressed.

Main concerns:

1. The decreased boundary response in pHPC with age is interpreted as evidence of hippocampal involvement in segmentation and indication that changes in boundary perception may be driven by this region. However, as the authors mention, perception of boundaries tends to differ between younger and older adults, and in particular is less consistent among older individuals.

- a. First, the age range of the independent observers should roughly match the age range of the fMRI participants. Otherwise, any effect of age could be due to differences in the similarity of the participants' boundaries to those of the observers.

Thank you, this is a very good point. Although the original manuscript was not clear on this point, evidence about aging effects on segmentation agreement is mixed, and actually much of the evidence suggests that it is relatively intact in cognitively normal older adults. To follow up on this point, we had a sample of 14 older adult raters (mean age = 73.83 years, SD = 6.27) and a new sample of 14 younger adult raters (mean age = 20.15 years, SD = 2.58) identify event boundaries in the film used in the present study. This allowed us to identify whether rating distributions substantially differed between young and older subjects (the boundary ratings provided by Ben Yakov et al. are average boundaries across participants with at least two raters agreeing, whereas we wanted a full, matched dataset to compare across age groups). In sum, we did not observe evidence that older adults were not segmenting events in a way that was different than younger adults for this particular stimulus. Critically, the 12 boundary timepoints used in our regressor for fMRI analyses, were identified by over half of our older adult sample (i.e., the criterion for these 12 'maximal agreement' timepoints). On average, the boundaries we included in fMRI analyses were identified by older adults 86% of the time, as compared with 82% of young subjects. Moreover, comparing older adults with the new sample of younger adults we collected demonstrates very high overlap across age groups in boundary ratings (see below). Together, we believe this is fairly compelling evidence that, at least for the film used in the present study, the timepoints we are modeling in this dataset were likely to be associated with subjective event boundaries in both older and younger adults.

b. However, even if the observers are age-matched, the main finding regarding older adults is of reduced consistency in their segmentation. This means there is a very plausible alternative interpretation of the results – that the older adults are more varied in where their boundaries are, leading to reduced hippocampal activity at group-defined boundaries. If the participants themselves had segmented the movie, there may be no decline with age in the same dataset. While (a) can be relatively easily addressed by collecting segmentations from older observers, I do not see how the second confound can be accounted for in the current dataset, and it has significant bearing on the ability to interpret the main finding (as well as the correlation with memory). I am happy to be proven wrong on this and would welcome the authors' thoughts.

The Reviewer's point is that, older subjects in the CamCAN sample might have shown normal boundary evoked activation, and the observed reductions might have been driven by inconsistencies in the locations of perceived event boundaries in older subjects. As noted above, data from previous studies is mixed as to whether segmentation agreement is reduced in older subjects. It is likely that the extent to which this occurs may be stimulus-dependent. Given the results from our new comparison of boundary ratings in older and younger adult samples for the movie used in the present study, it seems unlikely that older subjects in the CamCAN study showed substantial reductions in segmentation agreement. However, there is another key point that argues against the possibility that the age-related changes observed here are solely attributable to increased variability in boundary points in older subjects. If we simply missed critical event boundaries in older subjects due to reduced segmentation agreement, we would expect a uniform age-related reduction in activation for every region that showed an overall activity increase at event boundaries. In fact, we found that activation in some posterior medial regions decreased with age, but activity at other regions in the network (mPFC and MTG) showed *increasing* activity with age. These findings are not

consistent with the idea that older subjects simply segmented the film at different points than younger subjects. We highlight this point in an existing portion of the Results section as follows:

Finally, visualizing the data as a time-course of activity averaged across the modeled event boundaries, it is clear that the age-related decline in pHPC reflects a reduction in the amplitude of the hemodynamic response, rather than a fundamentally different shape, or a temporal shift of the response (Fig. 2A). Thus, age-related differences are not due to a failure to account for a shifted peak of the response. It is also clear that boundary-evoked response time courses and their relationship with age are less robust in aHPC than pHPC (Fig. 2B).

We also added an additional section to the Results to address this issue:

2.5. Older and younger participants segment the video stimulus similarly.

One possible explanation for age-related changes in boundary-evoked responses is inconsistency of perceived event boundaries in older adults. Though we cannot rule out that older participants in the CamCAN dataset simply 'missed' a number of the 12 event boundaries we modeled, we addressed this possibility in a separate sample of participants. We collected a sample of 14 older adults to provide some evidence as to whether there were age-associated differences in boundary detection. We found no evidence for this. In our smaller sample of older adults, at least half of the sample identified each of the 12 boundaries (i.e., the criterion for the 12 'maximal agreement' boundaries from the sample of young adults who provided the ratings), and the boundaries were overall identified by our sample 86% of the time (compared to 82% of the time in our younger adult sample). Thus, though it cannot be absolutely ruled out, it is unlikely that older adults simply perceived the key event boundaries differently than young adults.

2. When correlating the pHPC and ANG in different age groups, there is a lower correlation for older adults. This could arise from the lower response in these regions, as a lower response is linked to lower snr and can lead directly to lower correlations. It is difficult to interpret this finding without accounting for the amplitude differences between the groups.

Thank you for bringing up this important issue. Based on a close review of the data we have and our manuscript in light of this and other feedback, we ultimately have decided to remove the pHPC-ANG correlation from the paper. Though we think that this is an interesting result, it is one that is not central to the focus of the paper and main analyses. Thus, it does not lend itself to clear interpretation in light of our other findings, and moreover, it is subjected to issues related to amplitude variations shared across

regions that we cannot conclusively prove are not driven by SNR variability. Overall, we feel that omission of this analysis (and corresponding figure) make the overall scope and interpretability of our results clearer.

Minor concerns:

1. The Introduction should explicitly state that increased hippocampal activity at boundaries has already been identified on the same dataset. I am sure this is accidental, but the current phrasing is somewhat misleading, as this is described as one of the predictions of the study (lines 75-77), despite having already been demonstrated. Similarly, the authors may wish to mention in the section about boundaries and visual flow that HPC boundary-evoked responses have already been shown not to be driven by low-level visual changes in this dataset, although with a different method of assessing perceptual change.

Thank you, this was indeed not an intentional omission of details. We have revised this section to more explicitly credit the important prior work on the matter:

Introduction:

We predicted that activity evoked by event boundaries in the hippocampus and PM network regions – as previously seen in this dataset [10] – would be disrupted in the aging brain. We further predicted that boundary-evoked activity would relate to individuals' episodic memory performance.

Results:

We attempted to control for low-level visual information in our GLM by modeling edge pixels in the video (also similar to Ben Yakov and Henson [10]), and entering the proportion of edge pixels in each time point as a continuous regressor.

2. There seem to be at least two places where the references may have been confused:
a. In the Introduction, lines 45-48, is Chen et al. indeed the intended citation? The Chen 2017 study finds that hippocampal inter-subject correlation during scenes is linked to subsequent recall rather than boundary-evoked activity.
b. In line 425, are the citations in the sentence the correct ones?

Thank you for catching these errors, which seems to have been due to a glitch with our citation manager. These errors have been fixed.

3. In what sense did activity at boundaries in the pHPC 'uniquely predict' recognition memory (line 140)? Does this mean of all ROIs in the stepwise regression it was the only one to predict story recall? Did any of the cortical PM network regions also predict memory?

Thank you for the request for important clarification here. First and foremost, as described to another Reviewer, we have simplified our approach by switching from a stepwise regression model to a simpler OLS multiple regression. An advantage of this approach is that it does not de-emphasize ‘non-significant’ predictor variables so the reader has the opportunity to assess the ‘weaker’ relationships as well. Please find a description of the analysis here, from our revised Methods:

4.8. Multiple linear regression analysis

To simultaneously assess the predictive power of all predictor variables of interest in explaining variance in story memory, we fit multiple linear regression models (linear_model.LinearRegression in scikit-learn’s linear_model package) using an ordinary least squares approach. Briefly, this approach allows us to assess the relationship between variables of interest (e.g., pHPC activity and Logical Memory delayed recall) while also accounting for other important variables of interest (e.g., age or head motion). The model predicted Logical Memory performance immediately and at a delay, including all ROIs and neuropsychological test scores, as well as age and head motion (average framewise displacement) as predictor variables.

And here is a table of the results from the model:

Variable	Coefficient	Std. Error	t	p
Constant	0.3505	1.048	0.334	0.738
pHPC	3.0788	1.455	2.116	0.035*
aHPC	1.9977	1.782	1.121	0.263
PMC	-0.8259	1.808	-0.457	0.648
ANG	0.6639	2.124	0.313	0.755
PHC	0.6288	2.637	0.241	0.809
MTG	0.4054	1.494	0.271	0.786
mPFC	-0.6881	2.175	-0.316	0.752
EVC	-2.3315	1.430	-1.630	0.104
PRC	-0.0061	1.677	-0.004	0.997
TP	0.0957	0.808	0.118	0.906
AMY	-0.9888	1.583	-0.625	0.532

Motion	1.5441	1.578	0.978	0.328
Verbal Fluency	-0.0107	0.045	-0.237	0.812
Visuospatial	0.0302	0.041	0.744	0.457
Word Memory	0.0033	0.026	0.126	0.900
Log Mem Imm	0.8831	0.027	33.250	<0.001*
Age	-0.0194	0.007	-2.905	0.004*

In the full regression model, we in fact did find that pHPC activity at event boundaries was the only regressor of neural activity that was predictive of delayed recall. We have made this clearer in the revised manuscript:

In pHPC, activity at event boundaries predicted recall in the Logical Memory task stories both immediately ($r = 0.171$, $p = 6.07e^{-05}$) (Fig. 3A), and after a 20-minute delay ($r = 0.198$, $p = 3.33e^{-06}$) (Fig. 3B). This relationship was not specific to any age group, as older adults ($r = 0.244$, $p < 0.001$), middle aged ($r = 0.201$, $p = 0.006$) and younger adults ($r = 0.181$, $p = 0.014$) all featured a significant correlation. Though the strength of the correlation appears to increase with age, pairwise comparisons between the strength of these correlations (via Fisher z-transforms) did not reveal significant differences. In contrast, despite considerable statistical power, there were no other ROIs for which activity was predictive of story recall. Additionally, only a marginal correlation was observed between boundary-evoked activity in aHPC and Logical Memory performance at a delay that did not reach significance ($r = 0.077$, $p = 0.074$) (Fig. 3D), and no meaningful correlation between aHPC and immediate recall (Fig. 3C).

4. While the pHPC remains significant in explaining delayed recall memory after accounting for age, it is just-significant (especially as both immediate and delayed recall were tested in both pHPC and aHPC). While this is stated in the results, perhaps the description of this finding in the discussion could be toned down a bit. I would mention in the discussion as well that age was by far the primary explanatory variable, and that the relationship between pHPC and delayed recall was driven primarily by age, but remained significant. To further probe this finding, it may be interesting to test whether pHPC activity predicts delayed recall within each of the age groups.

Thank you, this is a fair observation. We have made this clarification in the Discussion:

Of the ROIs investigated here, pHPC activity at event boundaries uniquely predicted delayed recall performance on an entirely separate test of memory for complex narratives. Although age and performance on the corresponding immediate recall test predicted the majority of variance in delayed recall, pHPC

activity during event boundaries in the video still predicted significant variance in memory performance.

Additionally, we took the Reviewer's suggestion and explored differences in pHPC-recall correlations across age groups. Though the relationship was numerically larger in older adults ($r = 0.244$) than middle aged ($r = 0.201$) and younger adults ($r = 0.181$), none of these are significantly different than one another when directly contrasted (via a Fisher z-transforms). This has been added to the Results:

This relationship was not specific to any age group, as older adults ($r = 0.244$, $p < 0.001$), middle aged ($r = 0.201$, $p = 0.006$) and younger adults ($r = 0.181$, $p = 0.014$) all featured a significant correlation. Though the strength of the correlation appears to increase with age, pairwise comparisons between the strength of these correlations (via Fisher z-transforms) did not reveal significant differences.

5. In line 161, the authors mention there was no significant relationship of pHPC boundary-evoked activity with other neuropsychological tests. As evidence they mention all p-values are larger than 0.05. It would be preferable to give the minimal p-value of these tests, so the reader can assess whether some tests were related to pHPC activity but did not reach significance, or whether there was clearly no relationship.

Thank you for this suggestion – we agree that that reporting the smallest corresponding significance value is informative. This has now been added:

Surprisingly, we did not observe any significant relationships between pHPC boundary-evoked activity and any other neuropsychological test of interest, including composite tests of memory, verbal fluency, and visuospatial performance (all $p > 0.05$; minimal $p = 0.15$ for word memory).

6. Could the authors provide some rationale for the way in which they defined the boundaries?

Thank you for the request for important clarification here. Our boundary ratings were obtained from a sample of 16 participants whose data were gathered by Ben-Yakov and Henson (*J Neurosci*, 2018). Briefly, their instructions indicated participants should make a keypress when “one event (meaningful unit) ended and another began” based on event segmentation behavioral paradigms in, for example, Zacks et al., 2010. We now make this protocol clearer in the manuscript:

The GLM included a discrete regressor for human-labeled event boundaries, defined by a sample of 16 independent observers gathered by Ben Yakov and Henson [11]. Briefly, these event boundaries were gathered by participants who, when watching the 8-minute video stimulus, were simply instructed to press a key when one meaningful unit (i.e., an event) ended and another began, in line with prior studies [19-22]. We included boundary timepoints for which at least half of the participants from the sample agreed (within 5s of a group-measured

boundary time), and which were no closer than 6s to one another, totaling 12 event boundaries across the time series. The logic for this selection process was to include timepoints that we were confident featured what the majority of observers would perceive as an event boundary.

7. I found the edge pixel analysis a bit unclear. How were the edge pixels calculated? Why was this method chosen to assess visual flow? If I understand correctly, the calculation was the proportion of edge pixels in each frame (averaged within TRs) – wouldn't this predict high-frequency visual information within each time point, rather than visual change across time points?

The Reviewer is correct that this edge envelope most basically captures high-frequency information within each time point. This means that it would consequently be sensitive to differences between high and low-frequency timepoints, thus capturing some aspect of change across TRs. Based on the Reviewer's question, we have updated and clarified our description:

We attempted to control for low-level visual information in our GLM by modeling edge pixels in the video (also similar to Ben Yakov and Henson [10]), and entering the proportion of edge pixels in each time point as a continuous regressor. Edge pixels were calculated via a python routine which read in the video stimulus, split it into its constituent frames, and in an automated fashion performed edge-detection on each frame (using python package `opencv`). The proportion of edge pixels to total pixel count was calculated (`NumPy`) for each frame, and was output into a comma-separated value file. We then resampled frame-by-frame edge information to correspond to the temporal scale of the fMRI data by averaging across adjacent frames within the interval of each TR (2470ms). This resultant temporally-smoothed vector served as our estimate of low-level visual information in each timepoint of the video. This approximates an envelope of framewise high-frequency visual information in the video that is modeled independently of event boundaries.

8. In lines 482-484, adding the range of ages, in addition to the mean, would be informative.

Thank you, we have now included this information in the revised manuscript:

Participants were binned into three age ranges of equal size (N = 182 each): Young (mean age = 32.78, range = 18-44), Middle (mean age = 53.97, range = 44-65), and Older (mean age = 75.51, range = 65-88) groups.

9. The authors mention in lines 383-384 that all participants reported not having seen the movie. It would be more accurate to say 'most participants', as some of the participants did report having seen the movie.

The Reviewer is absolutely right. Apologies for this error, which has been corrected.

Reviewers' comments:

Reviewer #1 (Remarks to the Author):

The authors have done a fine job responding to my concerns

Reviewer #2 (Remarks to the Author):

Major concerns

Points 1 and 2:

The authors now address our concern about specificity of the effects to boundaries in their ROI analysis. It is great that the authors use boundary > within-event contrast for the correlation with age. We do think, however, and this might not have come across clearly in our previous review, that the authors should still report the correlations with boundary activity as well, potentially in a supplementary. From the perspective of a new reader that is not familiar with the previous version, the subtraction measure (while controlling for activation related to uninteresting factors) leaves open the question of whether the correlation stems from a reduction in the boundary condition across age or, potentially, an increase in activation in the within-event condition. As we know, the correlation indeed stems from the boundary condition, in line with the authors interpretation. Thus, it would be advisable to refer to it in the manuscript, and report that correlation as well in the manuscript or in a supplementary.

However, regarding the whole-brain analysis, the map showing different regions for boundary vs. baseline compared to within-event vs. baseline is nice as preliminary evidence. However, to argue for specificity (as the authors do in the beginning of this Results section), the authors should report the whole-brain contrast of boundary > within event. It could be that even though in these regions activation for boundary was significantly different than baseline, and was not so for within-event, activation in boundary may still not significantly differ from within event. Thus, as is, this analysis limits the current interpretation, and does not warrant the argument that these regions are specifically modulated by boundaries.

Point 3: The authors fully addressed our concern, and kudos for running these additional groups of participants.

Point 4: The authors fully addressed our concern.

The authors additionally addressed all of our minor concerns

I am supportive of publication after addressing the minor comments above. Regarding the whole-brain analysis, if no region is showing a significant effect of boundary > within-event, the authors could either omit the analysis altogether, or report that no region was detected since that is their main thesis and focus of the paper.

We thank the Editor and our Reviewers for their continued consideration of our manuscript. We are glad to have addressed the majority of the Reviewers' concerns, and appreciate Reviewer 2's additional comments. We agree with the points they raise, and address them in-line below. We hope that, with these improvements having been made, the manuscript is suitable for publication. Additionally, we have decided to make a minor change to certain terminology throughout the manuscript that we wish to make the Editor and Reviewers aware of. We changed the description of our Early Visual Cortex ROI to instead be Visual Cortex, given that this ROI includes a number of areas beyond primary visual cortex (regardless of the descriptor, we believe such regions offer a good contrast against our theoretically-motivated ROIs).

Reviewer #1 (Remarks to the Author):

The authors have done a fine job responding to my concerns

Reviewer #2 (Remarks to the Author):

Major concerns

Points 1 and 2:

The authors now address our concern about specificity of the effects to boundaries in their ROI analysis. It is great that the authors use boundary > within-event contrast for the correlation with age. We do think, however, and this might not have come across clearly in our previous review, that the authors should still report the correlations with boundary activity as well, potentially in a supplementary. From the perspective of a new reader that is not familiar with the previous version, the subtraction measure (while controlling for activation related to uninteresting factors) leaves open the question of whether the correlation stems from a reduction in the boundary condition across age or, potentially, an increase in activation in the within-event condition. As we know, the correlation indeed stems from the boundary condition, in line with the authors interpretation. Thus, it would be advisable to refer to it in the manuscript, and report that correlation as well in the manuscript or in a supplementary.

However, regarding the whole-brain analysis, the map showing different regions for boundary vs. baseline compared to within-event vs. baseline is nice as preliminary evidence. However, to argue for specificity (as the authors do in the beginning of this Results section), the authors should report the whole-brain contrast of boundary > within event. It could be that even though in these regions activation for boundary was significantly different than baseline, and was not so for within-event, activation in boundary may still not significantly differ from within event. Thus, as is, this analysis limits the current interpretation, and does not warrant the argument that these regions are specifically modulated by boundaries.

Thank you for fleshing your point out, and we apologize for not fully understanding and implementing the requested revision in the original set of comments. Apologies for that. We agree with the reviewer regarding the need to report both the raw boundary-evoked response values as well as those adjusted by the subtraction measure. This not only bolsters confidence in the basic findings, but more importantly, aids in clarity. We have implemented the reviewer's suggestion to make note of the more direct correlations in the results section, and to provide those plots in the supplementary materials, while mainly emphasizing the subtraction measure in the main manuscript.

Results, Pages 6-7:

To enhance the specificity of our estimates of activity at event boundaries, two steps were taken: (1) we modeled a regressor of non-interest modeling high-frequency visual information, (2) we modeled within-event timepoints equal to the number of boundary timepoints (see Methods for details), which we subtracted from boundary-evoked activity. This subtraction procedure addresses potential ambiguities in whether age-related changes in boundary-evoked activity are specific to event boundaries per se, or alternatively, a change in overall activity levels (i.e., during event boundaries, as well as within events). Our analyses here will thus be conducted over boundary-evoked activity minus within-event activity. However, analyses related to boundary-evoked activity alone, without this adjustment, can be seen in Supplementary Figures 1-4. We note that the basic findings reported below are unchanged as a function of this subtraction.

Below, we show the Supplementary Figures in question (please note the description in the caption, indicating these data are *not* adjusted for within-event activity as in the main text):

SUPPLEMENTARY FIGURES

Figure S1: Boundary-evoked activity and age-related decline in anterior and posterior hippocampus. Data are displayed as activity evoked by event boundaries, not adjusted for within-event activity as in the main text. (A) Activity at event boundaries significantly declines with age in pHPC. (B) This effect holds when grouping, and comparing across Young, Middle, and Older individuals. (C) Though a similar relationship is observed in aHPC, it does not reach statistical significance. (D) The Older adult group nonetheless shows significantly lower activity than the Young group. (* indicates a significant post-hoc comparison via Tukey's HSD.)

Figure S2: Boundary-evoked activity and age-related decline in anterior and posterior hippocampus. Data are displayed as activity evoked by event boundaries, not adjusted for within-event activity as in the main text. pHPC activity at event boundaries significantly predicts memory for stories in immediate (A) and delayed (B) recall conditions. These relationships did not reach significance in aHPC (C, D).

Figure S3: Declines and increases in boundary-evoked activity in the posterior-medial network. Data are displayed as activity evoked by event boundaries, not adjusted for within-event activity as in the main text. Age-related declines in boundary-evoked responses in angular gyrus (A,B), posterior-medial cortex (C,D), and parahippocampal cortex (E,F). Age-related increases in boundary-evoked responses in medial prefrontal cortex (G,H) and middle temporal

Figure S4: Anterior-temporal regions and early visual cortex do not show age-related changes. Data are displayed as activity evoked by event boundaries, not adjusted for within-event activity as in the main text. Boundary-evoked responses were not observed in (A,B) perirhinal cortex or (C,D) temporal poles. Boundary-evoked responses were observed in (E,F) early visual cortex, but these did not significantly change with age.

Methods, Page 27:

Thus, to control for within-event activity, we subtracted within-event betas from boundary-evoked betas as is displayed in the main figures. Analyses over boundary-evoked activity *not* featuring this subtraction are available in Supplementary Information. We note that, importantly, our basic findings are unchanged regardless of whether the within-event beta subtraction is performed.

Methods, Pages 28-29:

We additionally conducted a confirmatory voxelwise analysis to examine whole-brain maps of activity corresponding to event boundaries. Data were masked to the extent of brain voxels (but not masked exclusively to gray matter). In line with our central hypotheses, we conducted a direct contrast of boundary-evoked activity minus within-event activity. Voxels featuring significantly greater activity during event boundaries than within-event timepoints were plotted, false discovery rate corrected such that significance was defined as $q < 0.05$.

Additionally, the reviewer raises an excellent point about needing a direct comparison between boundary and within-event activity. As it stands, the reviewer is correct that this is inferred, but was not directly contrasted. We have now replaced the original Figure 7 (showing Event Boundary with Within-Event activity separately) with a new figure showing a direct voxelwise contrast, which can be seen below. We note that this activity map is fairly similar to the original Event Boundary-only map, though this makes sense given that this effect previously showed a significant effect against baseline with Within-Event timepoints also being simultaneously modeled:

2.4. Voxelwise analysis: boundary-evoked activity in the hippocampus and Posterior Medial network is distinct from within-event timepoints.

We next ran a confirmatory whole-brain voxelwise analysis to assess the selectivity of boundary-evoked responses to the PM network. This voxelwise analysis revealed that boundary-evoked activity versus within-event activity was centered on the hippocampus and PM network (FDR $q < 0.05$) (Fig. 7). This result consistent with our ROI-based findings described above, as well as with prior reports indicating BOLD activity in PM regions is sensitive to naturalistic event structure [9,10,27]. Taken together, this confirms that our hippocampal and PM network ROIs are reliably driven by event boundaries, which stands in contrast to activity observed within an event.

Figure 7: Voxelwise analysis of boundary-evoked activity versus within-event activity. Greater boundary-evoked activity than within-event activity is observed throughout the hippocampus and PM network, consistent with ROI-based analyses. Significance is evaluated at a corrected FDR threshold of $q < 0.05$.

Point 3: The authors fully addressed our concern, and kudos for running these additional groups of participants.

Point 4: The authors fully addressed our concern.

The authors additionally addressed all of our minor concerns

I am supportive of publication after addressing the minor comments above. Regarding the whole-brain analysis, if no region is showing a significant effect of boundary > within-event, the authors could either omit the analysis altogether, or report that no region was detected since that is their main thesis and focus of the paper.

Reviewer #2 (Remarks to the Author):

The revisions are satisfactory. No further requests or clarification needed.

***REVIEWERS' COMMENTS:

Reviewer #2 (Remarks to the Author):

The revisions are satisfactory. No further requests or clarification needed.

We are pleased to know that the Reviewer found our revisions addressed their concerns. We thank the Reviewer for their time and their important feedback.